# Intercellular network structure and regulatory motifs in the human hematopoietic system

Wenlian Qiao[1], Weijia Wang[1], Elisa Laurenti[2,3], Andrei L Turinsky[4], Shoshana J Wodak[4,5], Gary D Bader[3,6,7], John E Dick[2,3] & Peter W Zandstra[1,7,8,9,10,*]

## Abstract

The hematopoietic system is a distributed tissue that consists of functionally distinct cell types continuously produced through hematopoietic stem cell (HSC) differentiation. Combining genomic and phenotypic data with high-content experiments, we have built a directional cell–cell communication network between 12 cell types isolated from human umbilical cord blood. Network structure analysis revealed that ligand production is cell type dependent, whereas ligand binding is promiscuous. Consequently, additional control strategies such as cell frequency modulation and compartmentalization were needed to achieve specificity in HSC fate regulation. Incorporating the *in vitro* effects (quiescence, self-renewal, proliferation, or differentiation) of 27 HSC binding ligands into the topology of the cell–cell communication network allowed coding of cell type-dependent feedback regulation of HSC fate. Pathway enrichment analysis identified intracellular regulatory motifs enriched in these cell type- and ligand-coupled responses. This study uncovers cellular mechanisms of hematopoietic cell feedback in HSC fate regulation, provides insight into the design principles of the human hematopoietic system, and serves as a foundation for the analysis of intercellular regulation in multicellular systems.

**Keywords** feedback regulation; hematopoietic stem cell; intercellular signaling
**Subject Categories** Network Biology; Stem Cells
**Mol Syst Biol. (2014) 10: 741**

## Introduction

The hematopoietic system is a distributed tissue consisting of multiple phenotypically and functionally distinct cell types. Hematopoietic stem cells (HSCs), at the apex of the hematopoietic developmental hierarchy, populate and sustain the system through highly coordinated self-renewal and differentiation processes. Increasing evidence suggests that HSC fate decisions are regulated in part via feedback mechanisms including HSC autocrine signaling and paracrine signaling from differentiated hematopoietic cells (Csaszar *et al*, 2012; Heazlewood *et al*, 2013). However, the key signaling molecules and cell types involved and how multiple often competing feedback signals act to regulate HSC fate in a coordinated manner are poorly understood.

We previously used mathematical modeling and bioinformatic strategies to systematically characterize the role of feedback signaling in regulating human umbilical cord blood (UCB) HSC fate *in vitro* (Kirouac *et al*, 2009, 2010). We identified lineage-dependent stimulatory and inhibitory signals that constitute a dynamic and complex feedback signaling network for hematopoietic stem and progenitor cell (HSPC) proliferation. This led to the development of an effective culture system capable of expanding human UCB HSC by globally diluting inhibitory feedback signals (Csaszar *et al*, 2012), pointing to the relevance of the network that our modeling approach uncovered. However, how the feedback signaling network is organized and how HSCs sense and interpret the signals produced by different cell types remains to be elucidated.

Network analysis is a powerful approach to detect the design principles of many types of distributed systems. This strategy has been used to interpret ecological (Olesen *et al*, 2007), social (Apicella *et al*, 2012), financial (Vitali *et al*, 2011), and molecular (Jeong *et al*, 2001) systems, but has never been applied to cell–cell communication (CCC) networks. We hypothesized that mapping the hierarchical hematopoietic signaling network would provide insight into its regulatory structure and function, in particular how feedback mechanisms control HSC fate decisions. From a network structure perspective, we were particularly interested in understanding how network structures including modular (network division into

1    Institute of Biomaterials and Biomedical Engineering, University of Toronto, Toronto, ON, Canada
2    Princess Margaret Cancer Centre, University Health Network, Toronto, ON, Canada
3    Department of Molecular Genetics, University of Toronto, Toronto, ON, Canada
4    The Hospital for Sick Children, Toronto, ON, Canada
5    Department of Biochemistry, University of Toronto, Toronto, ON, Canada
6    Department of Computer Science, University of Toronto, Toronto, ON, Canada
7    The Donnelly Centre, University of Toronto, Toronto, ON, Canada
8    Department of Chemical Engineering and Applied Chemistry, University of Toronto, Toronto, ON, Canada
9    McEwen Centre for Regenerative Medicine, University of Health Network, Toronto, ON, Canada
10   Heart & Stroke/Richard Lewar Centre of Excellence, Toronto, ON, Canada
     *Corresponding author. Tel: +1 416 978 8888; E-mail: peter.zandstra@utoronto.ca

sub-networks) and promiscuous (overlapping connectivity and subspecialization of network components) strategies impact hematopoietic system behavior and HSC fate regulation.

Existing hematopoietic intercellular signaling networks have been constructed based on theoretical interactions between cells (Frankenstein *et al*, 2006) or curation of ligand–receptor interactions in heterogeneous cell populations (Kirouac *et al*, 2010). By taking advantage of high-resolution sorting of hematopoietic cells and transcriptome profiling, we created a CCC network to represent intercellular signaling between 12 highly resolved and phenotypically defined populations of stem, progenitor, and mature cell types from uncultured human UCB samples. We computationally analyzed the properties of the system and validated predictions using *in vitro* HSC fate responses to network-predicted HSC-targeting ligands. Our results support a model whereby differentiated hematopoietic cells influence HSC fates by regulating key intracellular regulatory nodes through cell type-dependent feedback signals. Control parameters such as relative cell frequency and local compartmentalization (niches) are opportunities to impose specificity in HSC fate regulation. Overall, our findings provide insight into the design principles of the human hematopoietic system focusing on the mechanisms of CCC in the feedback regulation of HSC fate. Further, our approach provides a fundamentally new strategy for analyzing intercellular regulation in multicellular systems.

# Results

### A hematopoietic cell–cell communication network is constructed from transcriptomic data

Our strategy for constructing and analyzing hematopoietic CCC networks is shown in Fig 1 that we will refer to throughout the manuscript. Transcriptomic data (Novershtern *et al*, 2011; Laurenti *et al*, 2013) of 12 phenotypically defined, highly enriched hematopoietic cell types (Fig 2A) were the resource for network construction (Fig 1; step 1a). The data captured the intuitive biological properties of corresponding cell types as defined by gene ontology (Fig 2B; see also Supplementary Table S1 and Materials and Methods). For example, stem and progenitor cells (hereafter collectively referred to as the primitive cells), except for megakaryocyte–erythroid progenitors (MEP), over-expressed HSC proliferation and differentiation genes; MEP and erythroblasts (EryB) over-expressed erythrocyte and megakaryocyte (Mega) differentiation genes; monocytes (Mono) over-expressed genes related to leukocyte and neutrophil (Neut) biological properties; and precursor B cells (PreB) over-expressed genes related to PreB differentiation.

To construct the CCC network, we compiled a database (Supplementary Table S2) of 341 receptors (or receptor genes) and their cognate ligands equivalent to 253 ligands (or ligand genes) (Materials and Methods). Hierarchical clustering of the receptor and ligand gene expression values recapitulated the developmental relationship (primitive cell compartment versus mature cell compartment) between the 12 cell types (Fig 2C), indicating similar expression of ligand and receptor genes in cells of the same developmental stage. Specifically, the primitive cells exhibited correlated receptor expression at higher confidence (average

$P = 0.005$) and correlated ligand expression at lower confidence (average $P = 0.175$) than the mature cells in which average $P$-values for receptor expression and ligand expression were 0.0900 and 0.0570, respectively. Thus, we suspected changes in the receptor and ligand expression in blood cells during progression through differentiation.

In the construction of CCC networks, we assumed that the differentially over-expressed genes of each cell type are predictive of the cell type's protein expression (Schwanhausser *et al*, 2011), and representative of the cell type's biological properties. To determine an appropriate false discovery rate (FDR) to define differential over-expression, we tested FDRs of 1%, 5%, 10%, 20% and 25% and then compared the set of receptors identified at each threshold to a benchmark of known cell type-associated receptors (see Materials and Methods). A FDR of 10% detected the known cell type-associated receptors with the optimal combination of sensitivity and specificity (Supplementary Fig S1), and thus the ligands (Supplementary Table S3A) and receptors (Supplementary Table S3B) differentially over-expressed according to this threshold were used in the subsequent analyses (Fig 1; step 1b).

A CCC network is a directional bipartite graph (Fig 2D) composed of connections between differentially over-expressed ligand and receptor genes of the cell types of interest, based on 933 ligand–receptor interaction pairs (Supplementary Table S2) involving the 341 receptors and 253 ligands in Fig 2C (Materials and Methods for network construction). Sixteen class-1 cytokines including CNTF, CSF2, CTF1, IL2, IL3, IL4, IL5, IL6, IL7, IL9, IL11, IL13, IL15, IL21, LIF, and OSM require interaction with hetero-multimeric receptors to initiate intracellular signaling cascades (Robb, 2007). Given that our network was constructed from gene expression data, from a modeling perspective, we assumed that the greater the number of receptor species that a cell expresses for a ligand, the higher the probability that the ligand binds to the cell. We considered the interactions of each ligand and its cognate receptors independently; this practice did not affect our conclusions on network structures as shown below. Some differentially over-expressed ligands and receptors did not have interaction partners in the analyzed cell types. For example, KIT expressed on HSC-enriched cells (HSCe: human UCB Lin$^-$CD34$^+$CD38$^-$CD45RA$^-$CD49f$^+$CD90$^{+/-}$) binds to SCF, a ligand produced by perivascular cells in the bone marrow niche (Ding *et al*, 2012), which our system did not have information about. Such ligands or receptors were connected to a hypothetical "Others" population representing an unknown number of additional cell types that potentially impact hematopoiesis. Based on these rules, a CCC network containing 1,344 ligand production-binding relationships between 249 ligand nodes and 13 cell nodes was constructed (Supplementary Table S4), of which 178 ligands mediated the connection between the 12 cell nodes of interest and 117 ligands targeting HSCe (Fig 1; step 1c). This CCC network paves a new way of depicting the hematopoietic hierarchy, and we next sought to analyze its properties.

As a starting point for our analysis, we separated the CCC network into two networks representing ligand production and ligand binding, respectively. The cell types were ranked in different orders based on the number of their interacting ligands in the two processes (Fig 2E). Distribution of the cell types based on the numbers of their produced ligands was approximated by a linear

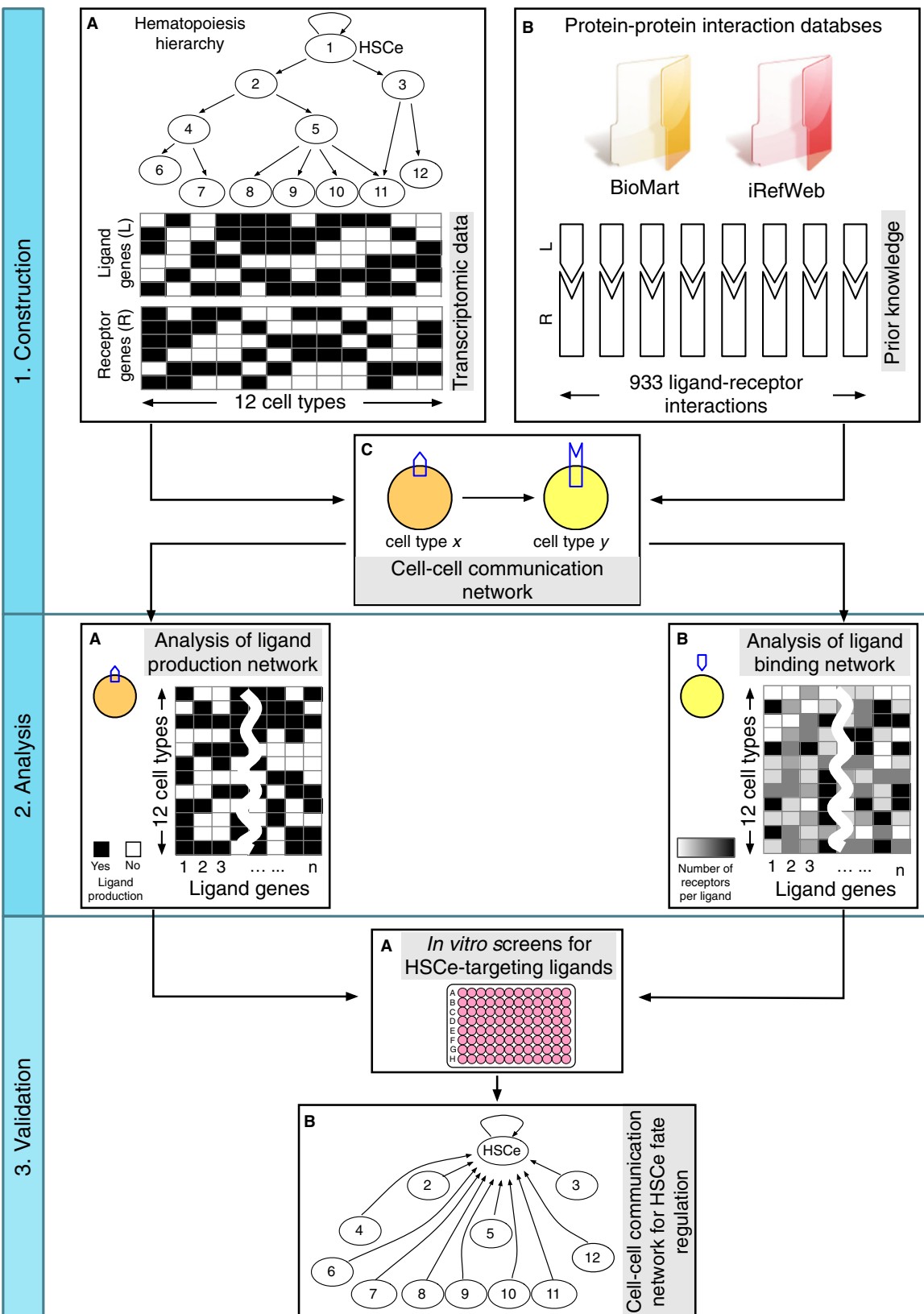

**Figure 1. Computational and experimental workflow of the study.**

The study is divided into network construction, analysis, and validation stages. Successive steps within each stage were alphabetically labeled. HSCe: human UCB HSC-enriched (Lin⁻CD34⁺CD38⁻CD45RA⁻CD49f⁺CD90⁺/⁻) cells.

Figure 2.

◀

**Figure 2.  Construction of cell–cell communication networks.**

A   Transcriptomic profiles of 12 phenotypically defined hematopoietic cell types isolated from human UCB were used. CMP, common myeloid progenitors; MEP, megakaryocyte–erythroid progenitors; GMP, granulocyte–monocyte progenitors; EryB, erythroblasts; Mega, megakaryocytes; Neut, neutrophils; Baso, basophils; Eos, eosinophils; Mono, monocytes; MLP, multilymphoid progenitors; PreB, precursor B cells.

B   Hematopoietic gene ontology enrichment analysis. Shown is the enriched gene ontology with hypergeometric (HG) *Z*-scores > 1.15.

C   Hierarchical relationships between the 12 cell types based on their ligand and receptor gene expression profiles. Hierarchical clusters for (i) 253 ligand genes and (ii) 341 receptor genes. Bootstrapped *P*-values (or approximately unbiased *P*-values) on the dendrograms score the uncertainty of the clusters. Dendrograms of gene clusters are not shown.

D   Concepts of cell–cell communication network constructed from differentially over-expressed ligand and receptor genes of each cell type.

E   Ranks of the 13 cell types including "Others" based on the numbers of their produced ligands and the numbers of their bound ligands.

See also Supplementary Fig S1.

function, whereas that based on the numbers of bound ligands was approximated by a step-like function—on average, EryB, Neut, and HSCe bound three times as many ligands as the other cell types. This difference posed the hypothesis that cells and ligands possess distinct interaction patterns in ligand production and binding processes, a hypothesis we explored by analyzing the structure of the two networks independently.

### Interaction between blood cells and ligands in the ligand production process is modular

A cell-to-ligand interaction, $A_{ij}$, in the ligand production network was defined if cell $i$ produced ligand $j$. Simultaneously, clustering the cell types and the ligands suggested that groups of ligands were associated with subsets of cells in the network (Fig 3A). Silhouette widths (Rousseeuw, 1987) measuring the relatedness of the cell types' ligand production supported the existence of 4 ligand–cell modules (Fig 3B, Supplementary Fig S2): the primitive cell module (HSCe + MLP + CMP + MEP + GMP), neutrophil–monocyte module (Neut + Mono), erythroid module (EryB), and a module of all the other cell types (Boso + Eos + Mega + PreB) (Fig 1; step 2a). *A priori* biological processes of 190 ligands (Supplementary Table S5) suggested that each blood cell module produced ligands with biased biological functions. For instance, ligands of the neutrophil–monocyte module enriched in exogeneous signals that inhibit cell survival (HG *Z*-scores were 1.63 and 2.98 for Mono and Neut, respectively) and signals that mediate cell survival via NF-κB (HG *Z*-scores were 2.15 and 1.43 for Mono and Neut, respectively); ligands of Baso, Eos, and PreB within the (Boso + Eos + Mega + PreB) module enriched in signals that direct differentiation cell fates of T helper cells (HG *Z*-scores were 1.17, 2.65, and 3.18 for Baso, Eos, and PreB, respectively); and ligands of EryB enriched in signals that regulate G1-S cell cycle transition (HG *Z*-score = 1.41) (Fig 3C). See Supplementary Table S6 for the other HG enrichment *Z*-scores.

In summary, our analysis suggested that blood cell ligand production is peculiar to blood cell identities, and a modular interaction structure exists in the ligand production network. This conclusion is robust to the choice of FDR threshold for differential gene over-expression (Supplementary Fig S2B) and the incorporation of hetero-multimeric receptor expression in network construction (Supplementary Fig S2C). Furthermore, ligand production of hematopoietic cell modules indicated characteristic biological properties. Considering HSC feedback regulation, this raised the possibility of HSC feedback control by cell module- or cell type-specific signaling.

### Interaction between ligands and blood cells in the ligand binding process is promiscuous

We next sought to determine whether the ligand binding network had a similar structure to the ligand production network. A ligand-to-cell interaction, $B_{ji}$, in the ligand binding network was defined if cell $i$ expressed receptor(s) for ligand $j$. Interrogation of the network (Fig 4A) using spectral co-clustering (Dhillon, 2001) suggested a significantly less modular interaction structure than in the ligand production network (Fig 3A) (*t*-test *P* < 0.001), with ubiquitously shared ligand binding among the 12 cell types due to non-specific ligand–receptor interactions (Supplementary Fig S3A). The promiscuous network structure is robust to the choice of FDR threshold for differential gene over-expression (Supplementary Fig S3B) and the incorporation of hetero-multimeric receptor expression in network construction (Supplementary Fig S3C). Interestingly, HSCe which normally reside in the bone morrow niche with progenitor and maturing cells (Fig 4B) interacted with ligands of the greatest diversity. This raised the question of how HSCe fate can be specifically regulated in response to physiological demand. We hypothesized two different mechanisms: relative cell frequency that allows more abundant cell types skew the ligand species and resources available to HSCe, and cell compartmentalization that limits the access of HSCe to locally available ligands. We then explored, computationally, the effects of the two mechanisms on the quantity and identity of HSCe-targeting ligands (Fig 1; step 2b).

To explore the role of cell frequency in skewing HSCe-targeting ligands, we compared ligand binding in two scenarios by assuming that the probability of binding a ligand is a function of cell frequency given non-regulated receptor ligand affinities. In the first scenario, we modeled ligand binding in the system of mono-nucleated cells (MNC) isolated from fresh human UCB samples. Based on flow cytometry analysis, Neut was the most abundant cell type in the system (Fig 4Ci) according to the phenotypic definition we used; consequently, the cell type was the major ligand sink that significantly influenced ligand accessibility of the other cell types (Fig 4Cii). In contrast, HSCe, a quantitatively underrepresented cell type in the MNC system, had negligible ligand access despite the large number ligands targeting the cell type (Fig 4A). In the second scenario, we modeled ligand binding using cell frequencies from progenitor cell-enriched UCB samples (Fig 4Di), in which cell composition is reminiscent of the progenitor enrichment seen during development or in the bone marrow niche (Nombela-Arrieta *et al*, 2013). Increased frequency of HSCe elevated their access to the available ligand resources (Fig 4Dii). This analysis indicates that controlling

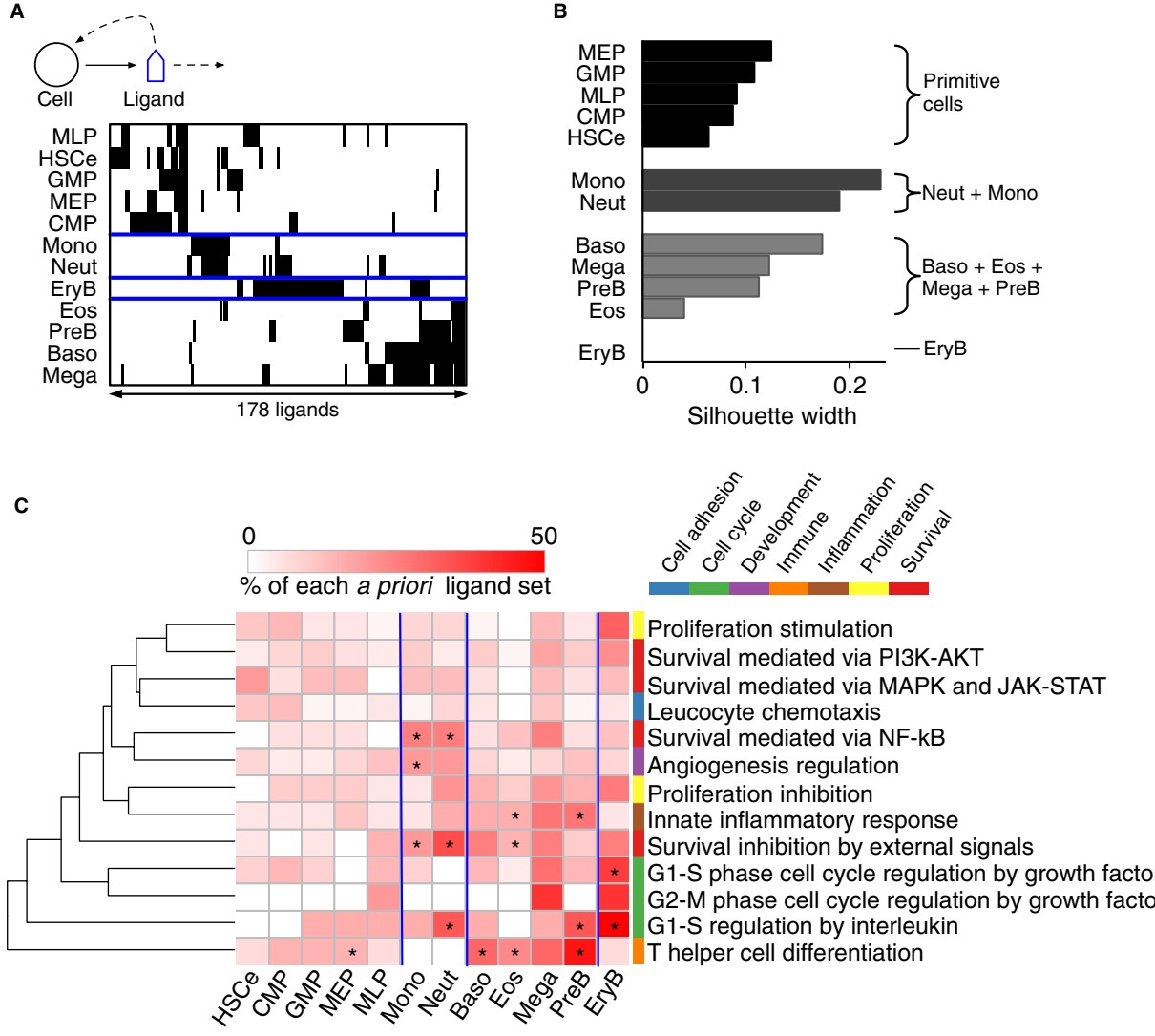

**Figure 3.  Modular ligand–cell interaction structure in the ligand production network.**

A   Hierarchical clustering based on Jaccard distances identifies four cell modules separated by the blue lines.

B   Silhouette widths for the four cell modules in (A).

C   Expression of *a priori* biological function-associated ligands by each cell module in (B). Asterisks (*) indicate the enriched ligand sets defined as HG Z-score > 1.15.

See also Supplementary Table S5 and Supplementary Figure S2.

hematopoietic cell relative frequency can modulate ligand exposure to HSCe.

Then, we explored the role of cell compartmentalization. While an increasing number of hematopoietic cell types such as erythroblasts (Soni *et al*, 2008), megakaryocytes (Huang & Cantor, 2009), monocytes (Chow *et al*, 2011), and B cell progenitors (Nagasawa, 2007) are found in the stem cell niche within the bone marrow environment, the exact location and direct feedback role of these cell types on HSC fate decisions is not clear. We used OR gates to model the feedback effect of these cell types on HSCe as a function of their localization based on the extant knowledge of 190 ligands (Supplementary Table S5). The model consisted of four compartments to represent cells of different developmental stages: HSCe themselves, progenitor cells (PC = CMP + GMP + MEP + MLP), mature cells in the stem cell niche (MCN = EryB + Mega + Mono + PreB), and granulocytic mature cells

in the peripheral blood or tissues (MCP = Baso + Eos + Neut) (Fig 4E). The spatial relationship between each compartment and HSCe was modeled by the probability of the ligands produced by the compartment reaching HSCe (Materials and Methods). Specifically, we assumed that (i) there is no diffusion for HSCe autocrine ligands, so the probability of HSCe autocrine binding $P_{HSCe}$ is 1; (ii) PC reside close to HSCe, so $P_{PC}$ is 0.8; (iii) MCN reside further away from HSCe than PC, so $P_{MCN}$ is 0.7; (iv) physical barriers between the stem cell niche and the peripheral tissues prevent MCP ligands from reaching HSCe, so $P_{MCP}$ is 0.1. We found that HSCe expressed a broad spectrum of autocrine signals including those thought to be important for HSC self-maintenance, whereas PC and MCN were the major producers of non-HSC supportive signals (Fig 4F).

*In vivo* monocytes, megakaryocytes, erythroblasts, and pre-B cells are primed to transit from the bone marrow to the peripheral blood.

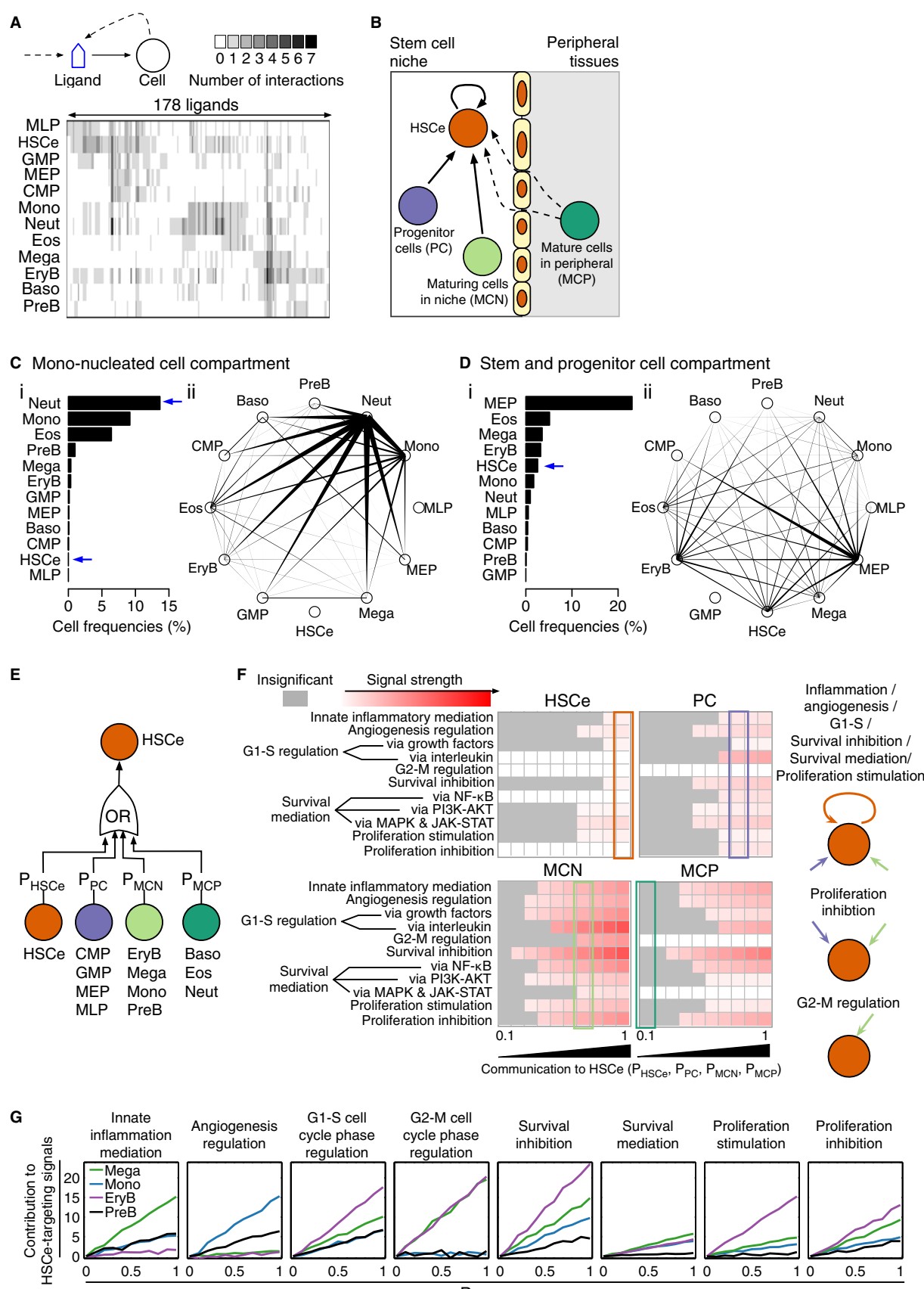

**Figure 4.**

This cell movement potentially alters the HSC microenvironment. We next sought to predict the spatial effect of Mono, Mega, EryB, and PreB on HSCe feedback regulation. Our simulation results (Fig 4G) revealed the importance of Mega-produced HSCe-targeting ligands in innate inflammatory response terms and the importance of Mono-produced HSCe-targeting ligands in regulating angiogenesis-associated terms. Strikingly, it was evident that EryB-produced HSCe-targeting ligands are associated with regulating cell cycle progression, cell survival and proliferation, which warrants future experimental validation. This analysis indicates that regulation of cell identities in HSCe microenvironment or niche can modulate ligand exposure to HSCe.

In summary, our analysis uncovered promiscuous ligand-to-cell interactions in the ligand binding network. HSCe were found to express receptors for a broad range of ligands, implying the existence of physical parameters such as relative cell frequency and compartmentalization in HSC fate regulation. Our subsequent simulation revealed a potential importance of Mega, Mono, and EryB ligands in HSC fate regulation. To explore how hematopoietic cell type-dependent signals feedback to HSCe, we next performed high-content *in vitro* experiments for HSCe-targeting ligands.

**Validation of HSCe-targeting ligands using a high-content *in vitro* phenotypic assay**

High-content *in vitro* experiments were performed by following the protocol in Fig 5A. HSCe-targeting ligands in the CCC network (Supplementary Table S4) were ranked according to the molecular interaction confidence scores (Ceol *et al*, 2010) for ligand–receptor interactions (Supplementary Table S2) and the receptor gene expression levels in HSCe from the Transcriptomic data. Thirty-three ligands were prioritized for experimental tests (Materials and Methods, Supplementary Table S7). We examined the phenotypic impact of each ligand on 40 HSC-enriched cells (HSC-e: Lin$^-$CD34$^+$ Rho$^{low}$CD38$^-$CD45RA$^-$CD49f$^+$) isolated from human UCB samples; this population contains approximately one NOD-*scid-IL2Rgc*$^{-/-}$ repopulating cell per 13 cells (combination of 1:10 for CD49f$^+$CD90$^+$ and 1:20 for CD49f$^+$CD90$^-$ HSC-enriched cells) (Notta *et al*, 2011). Each ligand was tested in a short-term assay at three doses in the presence of three basal cytokines (BC)—SCF, THPO, and FLT3LG (Petzer *et al*, 1996; Madlambayan *et al*,

2005; Csaszar *et al*, 2012). On day 7, the numbers of CD34$^+$CD133$^+$CD90$^+$ cells (defined as HSC-enriched cells) (Mayani & Lansdorp, 1994; Dorrell *et al*, 2000; Danet *et al*, 2001; Ito *et al*, 2010), CD34$^+$ cells that were CD133$^-$ or CD90$^-$ (defined as progenitor cells; see Supplementary Fig S5 for functional quantification using the colony-forming cell assay), and CD34$^-$ cells (defined as mature cells) were quantified. The BC cocktail-supplemented culture output 704 ± 425 (mean ± s.d. from 33 biological replicates) cells consisted of 6.35 ± 3.21% HSC-enriched cells, 27.75 ± 6.86% progenitor cells, and 65.90 ± 10.04% mature cells. This established a reference for detecting the effects of test ligands on HSC-e fate decisions (Supplementary Fig S6). In addition to the BC cocktail, TGFB1 (10 ng/ml) (Batard *et al*, 2000) and StemRegenin 1 (SR1, 0.75 μM) (Boitano *et al*, 2010) were used as the negative and positive control for HSC-e expansion, respectively (Fig 5B).

*In vitro* effect of the 33 ligands was quantified by signed one-tail *P*-values from the nested ANOVA detailed in the Materials and Methods (Supplementary Fig S7A). *P*-values of the 35 ligands (including TGFB1 and SR1) at their most effective dose on human UCB HSC-e are shown in Fig 5C. For ligands that did not have any significant effect, results of the highest working concentrations were reported. See Supplementary Fig S8 for cell number comparison between the tested conditions and the BC control. See Supplementary Tables S8 and S9 for results of all the testing conditions. These *in vitro* data allowed us to examine the impact of the cell types of interest on HSC fate regulation in the CCC network.

**Provisional feedback signaling networks for cell type-associable HSC fate modulation**

Measurement of the *in vitro* effect of the 33 ligands on HSC-e allowed creation of a directional CCC network. First, we categorized each ligand into one of the five functional categories [inducing quiescence, inducing self-renewal, inducing differentiation, inducing proliferation (self-renewal + differentiation), and inhibiting proliferation] in terms of their manipulation in HSC-e fate decisions using the *P*-values in Supplementary Table S9 and the classifier in Table 1. A representative ligand is given for each category in Supplementary Fig S7B. The ligands, at the working concentrations shown in Fig 5C, were categorized with different confidences (Fig 6A). Collectively, 27 out of the 33 ligands of interest were

---

**Figure 4. Promiscuous ligand–cell interaction structure in the ligand binding network.**

A  Spectral co-clustered adjacency matrix of ligand-to-cell interactions. The gray scale indicates the number of receptor genes expressed by a cell type for each of the 178 ligands.

B  Schematic *in vivo* HSCe feedback signaling network.

C  Cell frequency-dependent ligand binding network in the mono-nucleated cell compartment. (i) Composition of mono-nucleated cells isolated from fresh human UCB samples (*n* = 3). (ii) Potential of apparent competition (PAC) computed from the network weighted by the cell composition shown in (i). Along the edge connecting node *i* and node *j*, the width at node *i* indicates the competitiveness of node *i* to node *j* in terms of ligand binding.

D  Cell frequency-dependent ligand binding network in the stem and progenitor cell compartment. (i) Cell frequencies in lineage-depleted cells isolated from uncultured human UCB samples (*n* = 3). (ii) PAC computed from the network weighted by the cell composition shown in (i).

E  Logic gates used to model *in vivo* HSCe feedback signaling. The probability (*P*) of a cell compartment feeding signals to HSCe is inversely proportional to the distance between the cell compartment and HSCe.

F  Simulated functional effect of HSCe, PC, MCN, and MCP on HSCe as a function of feedback probability *P*. The color map indicates average signaling strength from 500 simulations. Insignificant cell–cell communication is colored in gray.

G  Simulated functional contribution of MCN cell type *x* (Mega, Mono, EryB, or PreB) to HSCe-targeting ligands as a function of the distance between MCN cell type *x* and HSCe. The simulation was performed at $P_{HSCe} = 1$, $P_{PC} = 0.8$, $P_{MCN-not\ x} = 0.7$, and $P_{MCP} = 0.1$. The magnitudes of contribution are with respect to $P_{MCN-x} = 0$, which is set to 0.

See also Supplementary Figure S3.

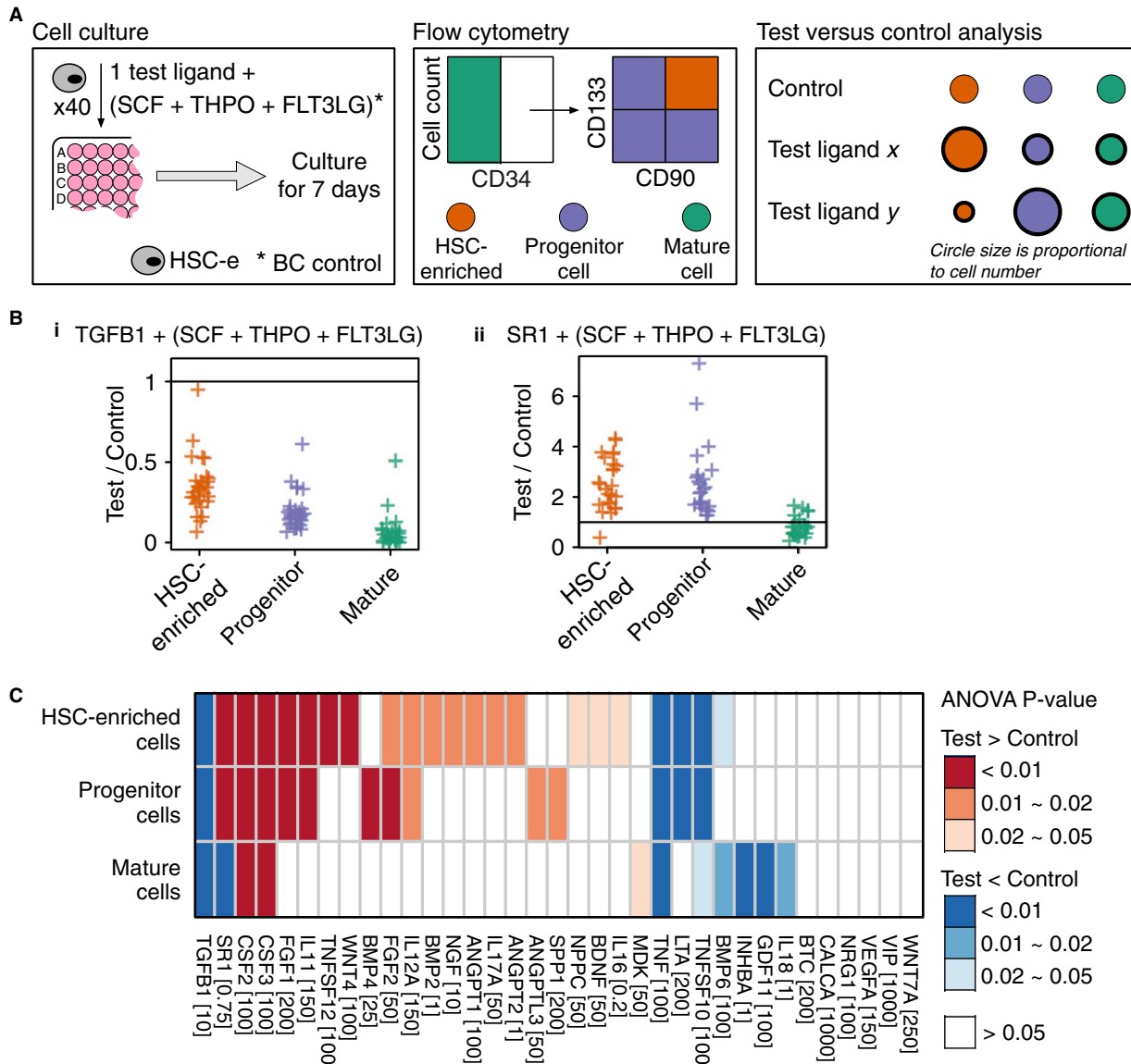

**Figure 5.  HSC-e respond to exogenously added HSCe-targeting ligands.**

A   The experimental and analytical protocol. HSC-e: human UCB Lin⁻RhoᶫᵒʷCD34⁺CD38⁻CD45RA⁻CD49f⁺. BC basal cocktail consisted of 100 ng/ml SCF, 50 ng/ml THPO, and 100 ng/ml FLT3LG.

B   Fold changes between the results of (i) negative control (TGFB1)/(ii) positive control (SR1) and that of the cell culture supplemented with BC only. HSC-enriched cells: CD34⁺CD133⁺CD90⁺; progenitors: CD34⁺ cells that are CD133⁻ or CD90⁻; mature cells: CD34⁻. Data are from 33 biological replicates.

C   Signed one-tail *P*-values from the nested ANOVA when comparing the cell counts of testing conditions to the BC control. Positive *P*-values indicate that effect of a test ligand was greater than that of the BC control, and negative *P*-values indicate the effect of a test ligand was less than that of the BC control. Ligand concentration is in ng/ml, except for SR1 that is in μM.

See also Supplementary Figs S4, S5 and S6.

found to direct HSC-e fate decisions (Fig 1; step 3a), indicating a significant enrichment of prediction capacity in this analysis (Binomial *P* = 0.0001, Materials and Methods).

Intriguingly, dose-dependent HSC-e fate regulation was observed for some ligands. For example, TNFSF10, at a working concentration of 1 ng/ml, did not affect the number of HSC-enriched cells, progenitor cells, or mature cells (ANOVA *P*-values were 0.2747, 0.2642, and 0.3721, respectively). When the ligand was used at 10 ng/ml, it led to an increase in the number of HSC-enriched cells

(ANOVA *P* = 0.0036), so it induced HSC-e self-renewal. At a working concentration of 100 ng/ml, however, the ligand led to a significant decrease in the number of HSC-enriched cells (ANOVA *P* = 0.0007), progenitor cells (ANOVA *P* = 0.0094), and mature cells (ANOVA *P* = 0.0207) (Supplementary Fig S6Bii), so it inhibited HSC-e proliferation, which may be due to the pro-apoptotic effect of the ligand (Zamai *et al*, 2000). Dose-dependent HSC-e fate regulation was also observed for FGF1, FGF2, IL11, and TNFSF12 (Supplementary Table S9). This result is reminiscent of differential

**Table 1. Functional definition of ligands for HSC-e fate regulation based on a cell number comparison between the conditions having the ligands of interest and the basal cytokine control.**

| | HSC-enriched cells | Progenitor cells | Mature cells |
|---|---|---|---|
| Neutral | – | – | – |
| Quiescence induction | – | – | ↓ |
| | – | ↓ | – |
| Self-renewal induction | ↑ | – | – |
| Differentiation induction | – | ↑ | – |
| | – | – | ↑ |
| | – | ↑ | ↑ |
| | ↓ | ↓ | ↑ |
| Proliferation induction | ↑ | ↑ | – |
| | ↑ | ↑ | ↓ |
| | ↑ | ↑ | ↑ |
| | ↑ | – | ↑ |
| | ↑ | ↓ | ↑ |
| Proliferation inhibition | ↓ | ↓ | ↓ |
| | ↓ | ↓ | – |
| | ↓ | – | ↓ |
| | ↓ | – | – |

Dash "–" indicates no change from the basal cytokine control.

activation of pathways that are involved in diverse biological processes (Kale, 2004). Furthermore, categorization of some ligands such as FGF2 (working concentration, WC = 50 ng/ml) and BMP6 (WC = 100 ng/ml) was sensitive to the statistical significance threshold, suggesting their indeterminate role in regulating HSC-e fate decisions may be context dependent. The ligands were excluded accordingly in the subsequent analyses.

We explored how ligands produced by different cell types influenced HSC-e fate decisions by performing a functional enrichment analysis for the ligands expressed by each of the 12 cell types in the CCC network using the ligand function categorization (Fig 6A) as a reference. To ensure that there were sufficient data to draw qualitative conclusions, the analysis was performed based on the categorization at the intermediate confidence level while excluding BMP6 in which categorization was indeterminate at that confidence level. Assuming each ligand acts independently in HSC-e fate regulation, this analysis allowed us, for the first time, to predict the role of each cell type in the HSC-e feedback regulation. As shown in Fig 6B, progenitor cells such as CMP, MEP, GMP, and MLP predominantly expressed ligands that induced HSC-e quiescence and self-renewal; EryB expressed ligands of diverse functions as expected from the results shown in Fig 3C. In contrast to a majority of the cell types, which expressed at most three types of directive signals for HSC-e fate decisions, HSCe expressed ligands inducing self-renewal, quiescence, and differentiation, and inhibiting proliferation. This is reminiscent of self-sufficient autocrine signaling of HSC (Kirito et al, 2005) possibly to compensate for their disadvantage in accessing exogenous signals

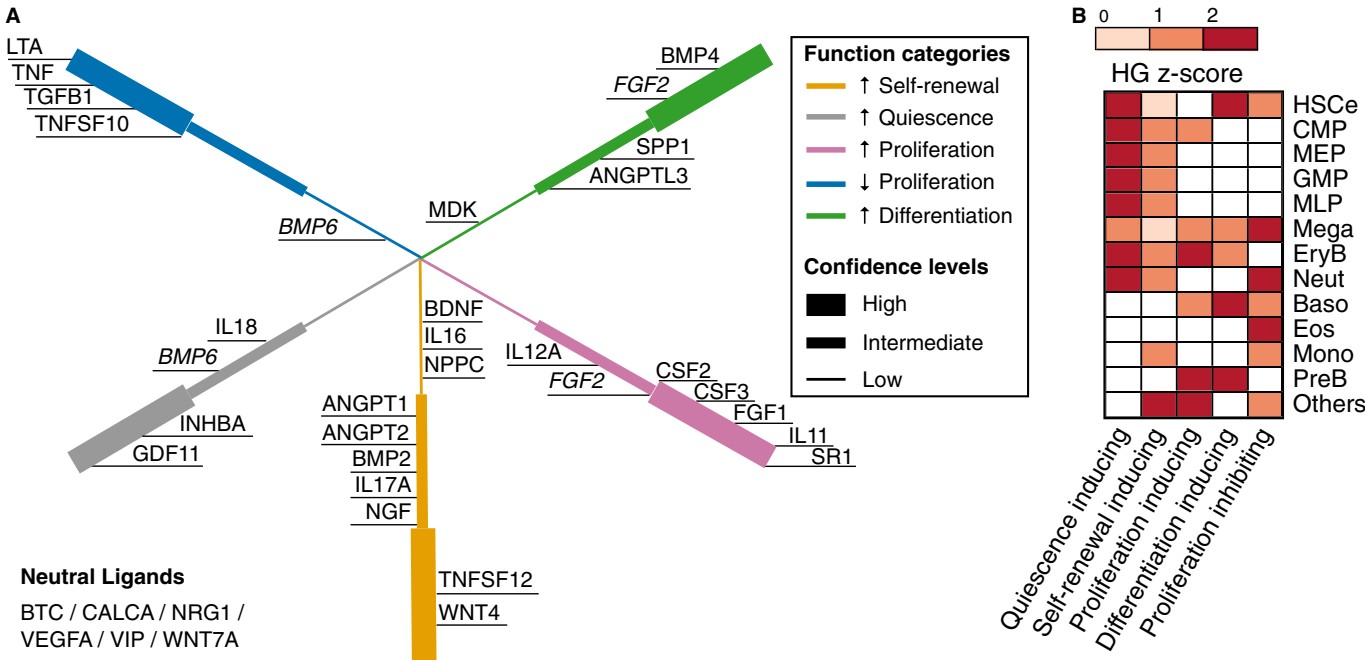

**Figure 6. In vitro experiments lead to functional categorization of HSCe-targeting ligands.**

A Functional categorization for the 35 HSCe-targeting ligands, including the negative control TGFB1 and the positive control SR1. The ligands were categorized at different confidence levels. High, intermediate, and low confidence levels refer to ANOVA P-value significance thresholds 0.01, 0.02 and 0.05, respectively. See Table 1 for definition of the functional categories.

B Functional enrichment was performed for the HSCe-targeting ligands produced by each cell type. The color scale indicates the HG enrichment Z-scores.

See also Supplementary Figs S7 and S8.

 

due to low cell frequency (Fig 4Ci). Collectively, we propose that both the progenitor cells and the mature cells regulate HSC-e fate decisions via feedback signaling yet through different mechanisms—the progenitor cells feed back HSC-e self-renewal and quiescence signals, whereas the more mature cells feed back HSC-e predominantly proliferation and differentiation signals (Fig 1; step 3b).

## Pathway enrichment analysis suggested intracellular regulatory motifs for HSC-e fate decision-making

The association between HSCe-targeting ligands and different cell types allowed us to construct a qualitative CCC network focusing on HSC-e fate regulation (Fig 7A). A database survey on the intracellular signaling pathways of the HSCe-targeting ligands suggested that intracellular regulatory motifs are associable with the ligands responsible for directive effects on HSC-e cell fate decisions *in vitro* (Fig 7B, Supplementary Fig S9, Materials and Methods). Specifically, signaling activity of the HSC-e quiescence-inducing ligands (such as BMP6 and IHNBA), self-renewal-inducing ligands (such as ANGPT1, ANGPT2, NGF, and TNFSF12), proliferation-inducing ligands (such as CSF2, CSF3, and IL11), and proliferation inhibitory ligands (such as TGFB1, TNFSF10, and TNF) were attributable to SMAD (permutation $P = 0.044$, Supplementary Fig S9A), NF-κB (permutation $P = 0.122$, Supplementary Fig S9C), STAT (permutation $P = 0.04$, Supplementary Fig S9C), and caspase cascade (permutation $P = 0.059$, Supplementary Fig S9D) pathways, respectively.

Our qualitative CCC network can be depicted in three ways: a directional network weighted by receptor frequency (Fig 7C), a directional network weighted by cell frequencies in the MNC compartment (Fig 7D), and a weighted directional network with compartmentalization (Fig 7E) overlaid to illustrate the roles of cellular dynamics and spatial distribution in HSC fate regulation through feedback signaling. For example, Neut was the largest cell population in the MNC isolated from human UCB (Fig 4C), so TNFSF10 and TNF from Neut were potentially the major signals to inhibit HSC-e proliferation. However, the stem cell niche-peripheral barrier would typically protect HSC-e from the inhibitory signals.

In summary, we combined the topology of the CCC network, the *in vitro* effect of 33 ligands on HSC-e fate decisions, and pathway information of the ligands. Our results support a model whereby hematopoietic cells influence HSC toward certain cell fates by regulating the key intracellular regulatory motifs through cell type-specific feedback signals.

## Discussion

While it is accepted that feedback regulation of HSC fate decisions is important to stable hematopoiesis (Csaszar *et al*, 2012; Heazlewood *et al*, 2013), it has been unclear how the feedback system operates. Extensive effort has been made to understand how stromal cells in the bone marrow microenvironment regulate HSC fate decisions (Zhang *et al*, 2003; Nakamura *et al*, 2010; Kunisaki *et al*, 2013). In addition, we propose a hematopoietic cell-driven feedback system that regulates HSC fate decisions through intercellular signaling.

We constructed a bipartite graph to represent the CCC network between 12 hematopoietic cell types isolated from human UCB (and

orphan signals entering the network). We found that the CCC network can be depicted in two formats based on signal directionality—ligand production and ligand binding, and each format was analyzed as an individual network. The high degree of modularity in the ligand production network pointed to cell type-specific production of ligands for HSC-e cell fate regulation. In contrast, the ligand-to-cell interactions in the ligand binding network were promiscuous, and HSCe were one of the cell types that bound the most ligands, suggesting that HSCe have broad environment sensing capacity (Takizawa *et al*, 2012). Our analysis raised important questions about how feedback specificity is achieved in HSC fate regulation. *In silico* simulation posed the hypothesis that additional control mechanisms including those observed *in vivo* (cell type frequency control and HSC niche localization or compartmentalization) are required to confer specificity in hematopoietic cell-mediated feedback regulation of HSC fate decisions. To test the hypothesis, we prioritized 33 HSCe-targeted ligands in the CCC network for *in vitro* experiments. We anticipated the roles of the 33 ligands in directing HSC-e fate decision using a cell surface marker expression-based phenotypic assay. The *in vitro* data allowed us to uncover what signals each of the 12 cell types feeds back to HSC-e. For instance, the mature cells, particularly Mono and granulocytes (Neut, Baso, and Eos), were found to express mainly inhibitory signals for HSC-e proliferation and inducing signals for HSC-e differentiation, which in combination can exhaust the HSC population because of the extensive cell cycling and division involved in the proliferation and differentiation processes (Hock *et al*, 2004; Zhang *et al*, 2006). However, under a normal *in vivo* condition, monocytes and granulocytes mainly circulate in the peripheral tissues; their secreted ligands have limited access to HSC in the bone marrow compartment because of the blood–bone marrow barrier. The identified importance of cell compartmentalization in protecting HSC from exogenous signals is consistent with our observation that global media dilution enhances *in vitro* HSC production when physical barriers between HSC and the mature cells are absent (Csaszar *et al*, 2012). We also found that progenitor cell types—CMP, MEP, GMP, and MLP—that typically co-localized with HSC in the bone marrow niche tend to function as a unit, enriched for ligands for HSC maintenance by inducing HSC quiescence and self-renewal. This finding supports the use of periodic primitive cell selection to increase *in vitro* HSC production (Madlambayan *et al*, 2005) and suggests technologies that target the HSC niche composition to control HSC fate *in vivo*.

The pathway enrichment analysis pointed to specific intracellular regulatory motifs associated with ligands of different *in vitro* effects on HSC-e fate. Specifically, HSC-e quiescence-inducing ligands such as BMP6 (Holien *et al*, 2012) and INHBA (Burdette *et al*, 2005) regulate the expression of SMADs to arrest cell growth. The HSC-e self-renewal-inducing ligands such as angiopoietins (Hughes *et al*, 2003), NGF (Descamps *et al*, 2001), and TNFSF12 (Kawakita *et al*, 2004) were found to regulate the activity of NF-κB in which deletion in the mouse hematopoietic system compromised the self-renewal and long-term hematopoietic repopulation ability of HSC (Zhao *et al*, 2012; Stein & Baldwin, 2013). The HSC-e proliferation-inducing ligands such as CSF2 (Carter, 2001; Gu *et al*, 2007), CSF3 (Harel-Bellan & Farrar, 1987), and IL11 (Yoshizaki *et al*, 2006) were found to induce the expression of STATs for cell proliferation. Finally, the HSC-e proliferation inhibitory ligands such as TGFB1 (Shima *et al*, 1999), TNF (Mallick *et al*, 2012), and TNFSF10 (Kischkel *et al*,

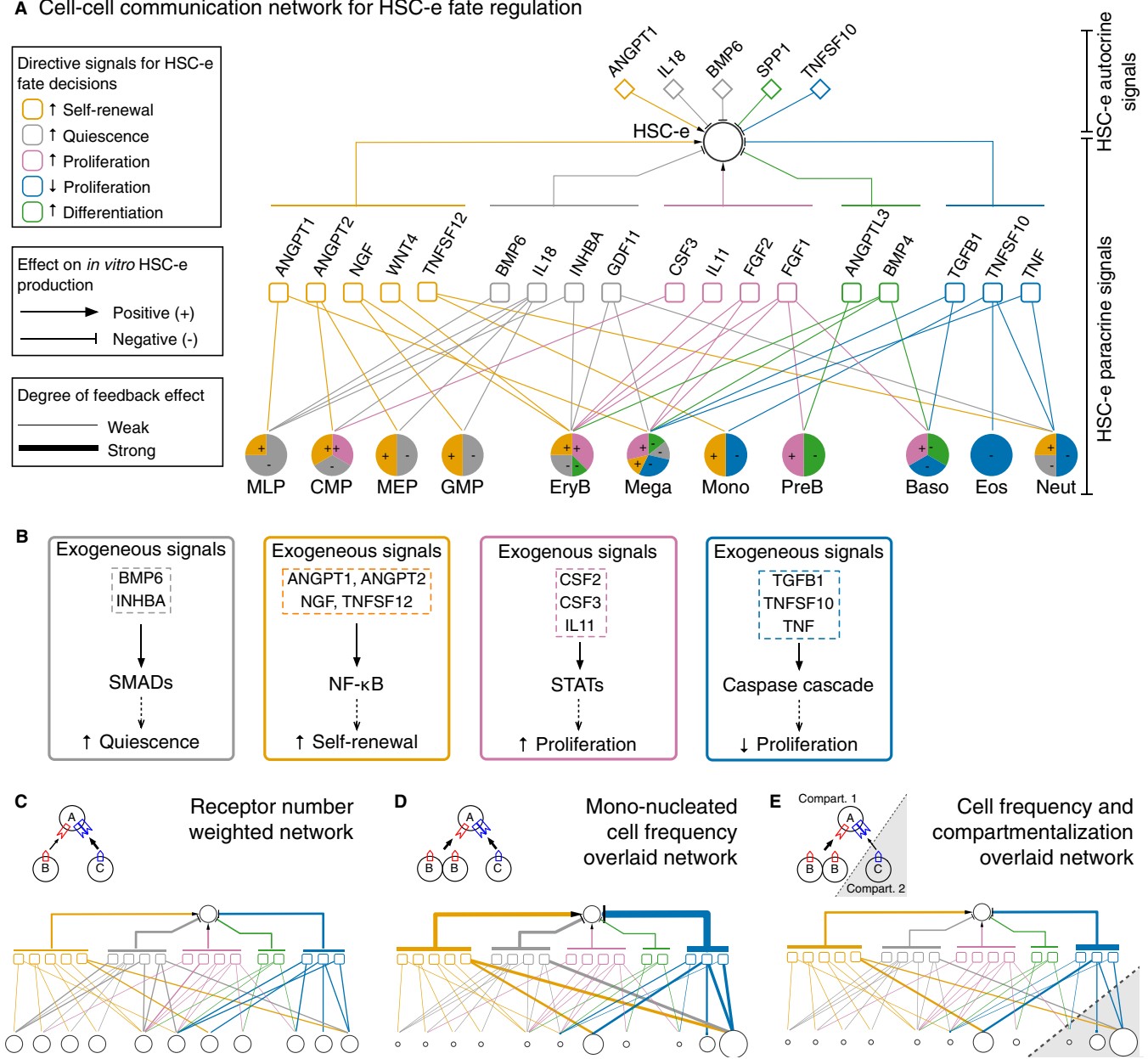

## A  Cell-cell communication network for HSC-e fate regulation

**Figure 7.  HSC-e feedback signaling network points to intracellular regulatory motifs for HSC-e fate regulation.**

A  Cell–cell communication network for HSC-e fate regulation. The hematopoietic cell-driven network for HSC-e fate regulation. The positive and negative feedback signals are in respect to *in vitro* expansion of CD34[+]CD133[+]CD90[+] cells.

B  Intracellular regulatory motifs associated with ligands of different directive effects on *in vitro* HSC-e fate.

C  Interactions between ligand-producing cells and ligands are weighted by the number of corresponding receptors (in terms of species) expressed in HSC-e. The thicker the edge, the higher the weight.

D  Interactions in (C) are weighted by cell frequencies obtained from fresh human UCB mono-nucleated cell samples shown in Fig 4Ci.

E  Interactions in (D) are weighted by spatial compartmentalization, where 10% of the ligands from peripheral compartment (Baso, Eos and Neut) reach HSC-e. The expressed ligands of the "Others" population, such as BMP2, LTA CSF2, and IL17, are not shown due to the lack of cell frequency information.

See also Supplementary Fig S9.

2000) initiated caspase cascade to cause cell death. Although many connections between exogenous ligand stimulation, pathway node activity, and cell phenotype changes were established in cancer cell lines, these connections led us to the anticipation that exogenous ligands direct HSC-e toward different cell fate by regulating the activity of specific cell fate decision-associated intracellular regulatory motifs, which opens opportunities for future study.

In summary, our results demonstrate the importance of cell-to-cell communication in human UCB stem cell fate control. Hematopoietic cells influence HSC toward certain cell fates by regulating

the key intracellular regulatory motifs through cell type-specific feedback signals. Further, control parameters such as relative cell frequency and spatial compartmentalization (niches) are opportunities to impose specificity in HSC fate regulation. A particularly interesting extension of our current work is to analyze how defects in HSC niche composition and physical structure or defects in HSC intracellular regulatory motifs affect feedback regulation of HSC fate decisions *in vivo* and consequently causes hematopoietic disorders such as leukemogenesis (Schepers *et al*, 2013).

One limitation of this study is that we used only transcriptomic data rather than proteomic data to construct the CCC network. Although there is a general agreement between mRNA and protein expression levels of ligands and receptors in mammalian cells (De Haan *et al*, 2003; Madlambayan *et al*, 2005; Schwanhausser *et al*, 2011), gaining better understanding of the dynamics of mRNA expression and the corresponding protein expression can be important in understanding context-specific network structures and their dynamic evolution. The newly developed mass cytometry (Bendall *et al*, 2011) offers a novel single cell proteomic approach to achieve this goal. A second limitation of this study is that we defined the exogenous effects of 33 ligands on HSC-e fate decision according to *in vitro* measurements of a cell surface marker expression-based phenotypic assay. Discrepancy between our observation about the *in vitro* effects of the tested ligands and their documented effects in literature may be attributable to the differences in experimenting cell populations and culture conditions. Further functional validation of the surface markers to cell function fidelity would certainly strengthen our analysis of network directionality; ultimately, our network should guide the selection of potentially novel HSC-e-regulating cell types, ligands, and their key intracellular signaling nodes for in-depth *in vivo* characterization. A final limitation of this study is that we used a static (human UCB) network to predict potentially dynamic feedback relationships between HSC-e and the other cell types. Exploring how the network connections change during culture evolution (Qiao *et al*, 2012) is an important next step. The assumption of our static network is direct (as opposed to indirect) feedback from each cell type to HSC-e. Although our *in vitro* study was specifically designed to enrich for direct effects of ligands on HSC-e by using the HSCe receptor expression information as a criterion for selecting test ligands and using a short culture time (7 days) (Csaszar *et al*, 2014), further analysis of multi-step and adaptive feedback is needed to strengthen links to *in vivo* hematopoiesis.

Collectively, cell–cell communication is fundamental to biologic tissues. However, it has not been extensively explored as a network because a large number of underpinning variables need to be considered. Here, we provide a framework to systematically depict cell–cell communication as a network while exploring the roles of cell frequency and spatial distribution in the system. As a next step, connecting the CCC network with more widely studied protein–protein interaction (Kirouac *et al*, 2010) and gene regulatory (McKinney-Freeman *et al*, 2012) networks through mechanistic models of intracellular signaling activity and the resulting cellular responses (Janes *et al*, 2005) will allow us to understand how HSCs integrate exogenous signals to make fateful decisions. The outcome will not only contribute to the development of more effective methods for HSC production, but also further our knowledge about HSC (niche) biology and cell–cell communication as a layer of biological regulation.

## Materials and Methods

### Microarray datasets

Illumina data of primitive cells and progenitor B cells (ProB: $CD34^+CD10^+CD19^+$; three biological replicates) were obtained from the authors of Laurenti *et al* (2013). The primitive cells are HSCe ($Lin^-CD34^+CD38^-CD49f^+CD45RA^-CD90^{+/-}$; 10 biological replicates), CMP ($Lin^-CD34^+CD38^+CD135^+CD45RA^-CD7^-CD10^-$; five biological replicates), MLP ($Lin^-CD34^+CD38^-CD90^-CD45RA^+$; five biological replicates), MEP ($Lin^-CD34^+CD38^+CD135^-CD45RA^-CD7^-CD10^-$; five biological replicates), and GMP ($Lin^-CD34^+CD38^+CD135^+CD45RA^+CD7^-CD10^-$; five biological replicates). The data are accessible at Gene Expression Omnibus (GEO) (Edgar *et al*, 2002) through accession number GSE42414. Quantile signals of the Illumina data were calculated using the normalizeQuantile() function in the limma package (v3.16.3) of BioConductor.

Affymetrix CEL files of mature cells and ProB ($CD34^+CD10^+CD19^+$; five biological replicates) were downloaded from GEO (accession number GSE24759 (Novershtern *et al*, 2011), accessed on 2011-11-20). The mature cells are Mega ($CD34^-CD41^+CD61^+CD45^-$; six biological replicates), EryB ($CD34^-CD71^-GlyA^+$; six biological replicates), Neut ($FSC^{hi}SSC^{hi}CD16^+CD11b^+$; four biological replicates), Baso ($FSC^{hi}SSC^{lo}CD22^+CD123^+CD33^{+/-}CD45^{dim}$; six biological replicates), Eos ($FSC^{hi}SSC^{lo}CD123^+CD33^{dim}$; five biological replicates), Mono ($FSC^{hi}SSC^{lo}CD14^+CD45^{dim}$; five biological replicates), and PreB ($CD34^-CD10^+CD19^+$; three biological replicates). Quality of the Affymetrix arrays was assessed using the simpleaffy (v2.32.0) and AnnotationDbi (v1.18.4) packages of BioConductor. The arrays with average background more than 2 s.d. from the mean background level of all arrays and the arrays with present percent is less than 1.5 s.d. from the mean present% of all arrays were not used for this study. Robust multi-array average (RMA) signals of the selected arrays were computed using the justRMA() function in the limma package (v3.16.3) of BioConductor. Affymetrix annotation for GeneChip U133AAofAv2 (GEO accession number: GPL4686) was used.

To combine the Illumina and the Affymetrix datasets, each dataset was normalized by the averaged gene expression signal of the respective ProB arrays. An averaged signal was calculated for probes of the same gene according to Entrez gene identifiers. The post-processed datasets were merged by Entrez gene identifiers.

### Ligand functional enrichment analysis

For the gene set enrichment analysis (GSEA) in Fig 2B, 13 hematopoietic gene sets (Supplementary Table S1) were compiled from the GeneGO database on 2012-11-15. GSEA was performed using the GSEA software (v2, http://www.broadinstitute.org) with the minimum gene set size equal to 1, and the other settings as defaults. See Supplementary Table S1 for GSEA *Z*-scores.

For the biological process enrichment analysis in Figs 3C and 4F, gene sets in Supplementary Table S5 were curated from the Meta-Core pathway database (http://thomsonreuters.com/metacore/, accessed on 2014-03-05). The material is reproduced under a license from Thomson Reuters; it may not be copied or re-distributed in

whole or in part without the written consent of the scientific business of Thomson Reuter.

### Ligand/receptor database

Using gene ontology terms "cytokine activity," "growth factor activity," "hormone activity," and "receptor activity," 417 genes with ligand activity and 1,723 genes with receptor activity were compiled from BioMart (Kasprzyk, 2011) (accessed on 2012-02-29). Ligand–receptor interaction pairs documented in public domains were compiled using the iRefWeb (Turner *et al*, 2010) resource (accessed on 2012-03-05). Additional 38 ligand–receptor interaction pairs from literatures (as on 2013-02-04) were included. See Supplementary Table S2 for the resulting 933 ligand–receptor interaction pairs.

### Hierarchical clustering

The hierarchical clusters in Fig 2C were obtained using the Ward agglomeration method with the Manhattan distance matrix. Confidence of the clusters was quantified by approximately unbiased (AU) *P*-values (Shimodaira, 2002, 2004), a type of bootstrap *P*-values, computed using the pvclust package (v1.2-2) in R (v3.0.0).

### Identification of differentially over-expressed genes

For the cell type of interest, one-way pairwise Wilcoxon test (R, v2.15.1) was performed between the gene expression profiles of the interested cell type and the profiles of each of the other cell types. *P*-values were adjusted using the Benjamini & Hochberg method (or false discovery rates, FDR). At a given threshold, the ligand and receptor genes that differentially over-expressed comparing to six other cell types (the threshold was set arbitrarily) were defined as the differentially over-expressed ligands and receptors of the cell type. The identified receptors of each cell type were compared to hematopoietic cell type-specific receptors using receiver operating characteristic (Supplementary Fig S1). The cell type-specific receptors are (1) ACVRL1 (for TGFB1), ENG (for TGFB1), EPOR (for KIT), FKBP1A (for TGFB1), IL2RG (for IL7), IL7R (for IL7), ITGAV (for TGFB1), ITGB6 (for TGFB1), ITGN8 (for TGFB1), KIT (for KITLG), LTBP1 (for TGFB1), LTBP4 (for TGFB1), MPL (for THPO), TGFBR1 (for TGFB1), TGFBR2 (for TGFB1), TGFBR3 (for TGFB1), VTN (for TGFB1), CD34 and ITGA6 (CD49f) for HSCe; (2) IL3RA (for IL3), CSF2RA (for CSF2), CSF2RB (for CSF2), CSF3R (for CSF2), EPOR (for KIT), KIT (for KIT), MPL (for THPO), CD34, CD38, FLT3 (CD135) for CMP; (3) MPL (for THPO), EPOR (for EPO), CD34 and CD38 for MEP; (4) CSF3R (for CSF3), CD34, CD38, FLT3, PTPRC (CD45RA) for GMP; (4) IL2RG (for IL7), IL7R (for IL7), CD34, PTPRC (CD45RA) for MLP; (5) MPL (for THPO), ITGA2B (CD41), ITGB3 (CD61) for Mega; (6) EPOR (for EPO), GYPA (CD235a) for EryB; (7) CD14 for Mono; (8) CD22 and IL3RA (CD123) for Baso; (9) IL3RA (CD123) for Eos; (10) FCGR3A (CD16) and ITGAM (CD11b) for Neut; and (11) IL2RG (for IL4), IL4R (for IL4), IL13RA1 (for IL4), MME (CD10), and CD19 for PreB.

### Network construction

Directionality of the CCC network was defined by the expression of ligand and receptor genes on the cell types of interest, and the

ligand–receptor pairs in Supplementary Table S2. If "Cell A" expresses a receptor for ligand *x* and "Cell B" expresses ligand *x*, an arrow is drawn from "Cell B" to "Cell A." Networks were built in R (v2.15.1) and visualized in Cytoscape (v2.8.3). The R code is available upon request.

### Bipartite network analysis

Clustering for the ligand production networks was performed based on Jaccard distances appropriate for binary graph adjacency matrices (Gower & Legendre, 1986). Clustering for the ligand binding networks was performed using the spectral co-clustering algorithm (downloaded from http://adios.tau.ac.il/SpectralCoClustering/ on 2013-06-01) appropriate for weighted graph adjacency matrices (Dhillon, 2001).

Potential of apparent competition (Muller *et al*, 1999) of cell type *i* to cell type *j*, $P_{ij}$, was computed as

$$P_{ij} = \sum_K \left( \frac{f_i R_{ik}}{\sum_I f_i R_{il}} \frac{f_j R_{jk}}{\sum_M f_m R_{mk}} \right),$$

where $f_i$ is the normalized cell frequency of cell type *i* by the total cell frequency of the analyzed cell types, thus $f_i$ is between 0 and 1; $R_{ik}$ is the number of receptors that cell type *i* expressed for ligand *k*; *K* is the total number of ligands that cell type *i* binds; *I* is the total number of ligands that cell type *i* binds; and *M* is the total number of cell types that ligand *k* binds. The figures were drawn by modifying the plotPAC() function in the bipartite package (v1.18) in R (v.3.0.0).

### Network comparison

To compare interaction patterns between the network of ligand source and the network of ligand sink, for each network, the numbers of overlapped ligands between one module and the other modules were obtained. The overlap of ligands between modules in the network of ligand source **S** = {9, 13, 10, 12, 12, 17}, and the overlap of ligands between modules in the network of ligand sink **T** = {75, 75, 69}. Two-sample *t*-test was performed for **S** and **T** in R (v3.0.0).

### Flow cytometry analysis

Human UCB samples were collected from consenting donors according to ethically approved procedures at Mt. Sinai Hospital (Toronto, ON, Canada). Mono-nucleated cells were obtained by depleting red blood cells (RBC) using RBC lysis buffer (0.15 M $NH_4Cl$, 0.01 M $KHCO_3$, 0.1 mM EDTA) as previously described (Kirouac *et al*, 2009). Lineage-negative (Lin⁻) cells were isolated from the mono-nucleated cell fraction using the StemSep system or the EasySep system for human progenitor cell enrichment (StemCell Technologies, Inc., Vancouver, BC, Canada), according to the manufacturer's protocol. Cell frequencies shown in Fig 4Ci and 4Di were obtained from mono-nucleated cells of fresh UCB samples and thawed Lin⁻ cell samples, respectively. The cells were stained using the following antibodies in 1:100 unless stated otherwise: CD90 (FITC, 1:50), CD38 (PE, PECy5, APC), CD45RA (1:50, APC), CD34

(PE-Cy7), CD49f (PE-Cy5, 1:50), CD7 (FITC), CD10 (FITC), CD135 (1:50, PE), CD45RA (1:50, APC), CD71 (FITC), CD235a (PE), CD61 (FITC), CD41 (PE), CD45 (PE-Cy7), CD14 (PE), CD16 (PE), CD11b (PE-Cy7), CD22 (FITC), CD33 (PE), CD123 (PE-Cy5), CD19 (FITC), and CD10 (PE). All the antibodies were from BD Biosciences, Mississauga, ON, Canada.

## Logic modeling

The effect of cell localization on the identity of HSCe-targeting ligands $\mathbf{M}_{HSCe}$ was simulated using an OR gate model:

$$\mathbf{M}_{HSCe} = (\mathbf{x}_{HSCe} \cdot \mathbf{L}_{HSCe}) \cup (\mathbf{x}_{PC} \cdot \mathbf{L}_{PC}) \cup (\mathbf{x}_{MCN} \cdot \mathbf{L}_{MCN}) \cup (\mathbf{x}_{MCP} \cdot \mathbf{L}_{MCP}),$$

where $\mathbf{L}_{HSCe}$, $\mathbf{L}_{PC}$, $\mathbf{L}_{MCN}$, and $\mathbf{L}_{MCP}$ are the differentially over-expressed ligands of HSCe, progenitor cells (CMP, GMP, MEP, and MLP), mature cells in the stem cell niche (MCN), and mature cells in the peripheral tissues (MCP). Randomly generated logic vectors $\mathbf{x}_{HSCe}$, $\mathbf{x}_{PC}$, $\mathbf{x}_{MCN}$, and $\mathbf{x}_{MCP}$ represented the probability ($P_{HSCe}$, $P_{PC}$, $P_{MCN}$, and $P_{MCP}$) of the ligands of each compartment to reach HSCe. Enrichment ($E$) of HSCe-targeting ligands $\mathbf{M}_{HSCe}$ in a biological process mediated by ligand set $\mathbf{B}$ was quantified as following:

$$E = \frac{n(\mathbf{M}_{HSCe} \wedge \mathbf{B})}{n(\mathbf{B})},$$

where $n(\mathbf{M}_{HSCe} \wedge \mathbf{B})$ is the number of HSCe-targeting ligands in biological process B, and $n(\mathbf{B})$ is the number of ligands in biological process B. For each test condition (i.e., combination of $P_{HSCe}$, $P_{PC}$, $P_{MNC}$, and $P_{MCP}$), enriched scores from 500 simulations were averaged. Content of 11 manually curated ligand sets of biological processes are tabulated in Supplementary Table S5.

## *In vitro* experiments

Human Lin⁻ cells were isolated from UCB samples collected from consenting donors according to ethically approved procedures at Mt. Sinai Hospital (Toronto, ON, Canada). Forty Lin⁻ Rho^low CD34⁺CD38⁻CD45RA⁻CD49f⁺ cells were sorted and dispensed per well in a 96-well V-bottom plate with a MoFloXDP flow cytometer (Beckman Coulter). The cells were cultured in a serum-free condition supplemented with 100 ng/ml SCF, 100 ng/ml FLT3LG, 50 ng/ml THPO, and a test ligand at specific concentration. On day 7, cells were stained. Total cell counts ($N_{Total}$), CD34⁺CD133⁺CD90⁺ cell counts ($N_{HSC-enriched}$), and CD34⁻ cell counts ($N_{Mature}$) were obtained using an LSRFortessa flow cytometer (BD Bioscience). Progenitor cell counts were calculated as $N_{Total} - N_{HSC-enriched} - N_{Mature}$. See also "optimization of *in vitro* experiments" in the Supplementary Information S1.

## Statistical analysis

To assess the effects of each test ligand (in addition to SCF, THPO and FLT3LG) on *in vitro* HSC-e fate decisions, a mixed-linear model was constructed with the experiment identifier as the random effect to account for the variability from experiment to experiment. The analysis was performed using the lme() function of the nlme

package (v3.1-113) in R (v2.15.1). The source code is provided as Supplementary Information S1.

Since we were mostly concerned with not missing any effective ligands (type II error) that will inform future research, nominal *P*-values of the mixed model were reported without correction for multiple tests. The ligands were categorized using definition in Table 1. Ligand categorization was performed for significance *P*-value thresholds of 0.01, 0.02 and 0.05 (Supplementary Table S9). See also "statistical analysis for *in vitro* experiments" in the Supplementary Information S1.

At the *P*-value threshold of 0.02, 5 ligands were found to be neutral to HSC-e and 27 were categorized into five functional categories (inducing HSC-e quiescence, self-renewal, differentiation and proliferation, and inhibiting HSC-e proliferation). Assuming the probability that a selected ligand is functional is 0.5 and that the effectiveness of test ligands was independent from each other, the ligand selection process was modeled as a binomial process with distribution X~B(33, 0.5), where 33 is the number of test ligands. The expected number of effective ligands was 33*0.5 ≈ 16. The probability of having 27 effective ligands is

$$P(X = 27) = \binom{33}{27} 0.5^{27}(1 - 0.5)^6 \approx 0.0001$$

Prior to the *in vitro* experiments for testing the activity of HSCe-targeting ligands on HSC-e, we sought to prioritize ligands for experiments. To do that, we performed a literature survey on ligands that had been used in *in vitro* cell culture of human cord blood-derived cells; 11 ligands fell in this category (Supplementary Table S7). Ligands such as ANGPT1, ANGPT2, ANGPTL3, and BMP2 had been used in mice or human bone marrow cells (Supplementary Table S7), so they were also prioritized for experiments in our study. Excluding these ligands from our analysis, 15 ligands out of 18 tested ligands were effective. The corresponding probability is

$$P(X = 27) = \binom{18}{15} 0.5^{15}(1 - 0.5)^3 \approx 0.003$$

To dictate the respective regulatory effects of HSCe, CMP, GMP, MEP, MLP, Mega, EryB, Mono, Neut, Eos, Baso, PreB, and Others on HSC-e cell fates, the tested ligands of each cell type were extracted from the CCC network in Supplementary Table S4. Functional enrichment analysis was performed for each cell type using hypergeometric *Z*-scores,

$$Z = \frac{k - n\frac{m}{N}}{\sqrt{n\frac{m}{N}\left(\frac{N-m}{N}\right)\left(\frac{N-n}{N-1}\right)}},$$

where $N = 117$ is the number of HSCe-targeting ligands expressed by the 13 cell types, $m$ is the number of ligands in a given function group, $n$ is the number of expressed ligands of the cell type of interest, and $k$ is the number of expressed ligands in the function group of interest.

## Functional HSC-e feedback signaling network

In Fig 7C, strength of the produced signals of function group $k$ from cell type $i$ to HSC-e was modeled as

$$S_{i,k} = f_i \sum_{n=1}^{N} R_{\text{HSC}-e,k,n},$$

where $f_i$ is the frequency of cell type $i$, $n$ is the number of expressed ligands of function group $k$ by cell type $i$, and R is the expression level of receptor gene $n$. Cell frequencies are from Fig 4Ci.

### Pathway analysis

Intracellular regulatory factors downstream of 16 out of the 19 ligands shown in Fig 7A are available in the MetaCore database (http://thomsonreuters.com/metacore/, accessed on 2014-04-01). The regulatory factors of each ligand were compiled and compared to the regulatory factors of the other ligands of the same functional group. Enrichment of ligands of the same functional group to each regulatory factor was calculated by a permutation test. For each regulatory factor, random functional categorization (quiescence induction, self-renewal induction, proliferation induction, and proliferation inhibition) was performed for 100,000 times. The ratio between the number of times that a regulatory factor randomly fell in a functional category more frequent than the actual categorization and 100,000 is defined as the permutation *P*-value. The results of pathway analysis for HSC-e differentiation-inducing ligands are not presented because pathway information was only found for one differentiation-inducing ligand BMP4, and the data are not sufficient for an enrichment analysis. The material from the MetaCore pathway database is reproduced under a license from Thomson Reuters.

**Supplementary information** for this article is available online: http://msb.embopress.org

### Acknowledgements

WQ was supported by Ontario Graduate Scholarships and a National Science and Engineering Research Council postgraduate scholarship. WW was supported by an Ontario Stem Cell Initiative post-doctoral fellowship. EL was supported by the Swiss National Science Foundation, Roche, and the FSBMB (Foundation Suisse pour les Bourses en Médecine et Biologie). This work was supported by grants to SJW from the Canadian Institutes of Health Research, the Ontario Research Fund, and the SickKids Foundation; grants to GDB from NRNB (U.S. National Institutes of Health, National Center for Research Resources Grant Number P41 GM103504); grants to JED from Genome Canada through the Ontario Genomics Institute, Ontario Institute for Cancer Research, and a Summit Award with funds from the province of Ontario, the Canadian Institutes for Health Research, a Canada Research Chair, the Princess Margaret Hospital Foundation, the Terry Fox Research Institute, Canadian Cancer Society Research Institute, and in part by the Ontario Ministry of Health and Long Term Care (OMOHLTC, the views expressed do not necessarily reflect those of the OMOHLTC); and grants to PWZ from the Human Frontier Science Program, the Leukemia and Lymphoma Society of Canada, the Canadian Stem Cell Network, and the Ministry of Research and Innovation of Ontario. PWZ is the Canada Research Chair in Stem Cell Bioengineering. The authors would like to thank the members of the PWZ laboratory and Dr. Daniel Kirouac for their helpful discussion.

### Author contributions

WQ and PWZ conceived and designed the study and wrote the manuscript. WQ performed *in silico* studies and analyzed *in vitro* data. WW performed *in vitro* experiments and contributed to drafting the manuscript. EL, ALT, SJW, GB and JED contributed reagents/materials/analysis tools. All the authors reviewed the manuscript.

### Conflict of interest

The authors declare that they have no conflict of interest.

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
