## [Review Process File · Molecular Systems Biology]

Intercellular Network Structure and Regulatory Motifs in the Human Hematopoietic System

Wenlian Qiao, Weijian Wang, Elisa Laurenti, Andrei L. Turinsky, Shoshana J. Wodak, Gary D. Bader, John E. Dick, Peter W. Zandstra

Corresponding author: Peter W. Zandstra, University of Toronto

Review timeline:

Submission date:	20 January 2014
Editorial Decision:	19 February 2014
Revision received:	25 April 2014
Editorial Decision:	28 May 2014
Revision received:	09 June 2014
Accepted:	17 June 2014

Editor: Maria Polychronidou

Transaction Report:

1st Editorial Decision

19 February 2014

Thank you again for submitting your work to Molecular Systems Biology. We have now heard back from the three referees who agreed to evaluate your manuscript. As you will see from the reports below, the referees acknowledge that the presented findings are potentially interesting. However, they raise a series of concerns, which should be carefully addressed in a revision of the manuscript.

Without repeating all the points listed below, among the more fundamental issues are the following:

- Additional analyses are required in order to convincingly validate the proposed cell-cell interaction network.

- Notably, the reviewers refer to the need to address the effect of the use of "literature anticipated HSC-targeting ligands" as one of the criteria for selecting ligands for the high-content screen, on the prediction capacity of the analysis.

- The role of heteromultimeric receptors in the regulation of the cells' responses needs to be taken into consideration.

- The additional experiments suggested by reviewer #2 and involving the functional validation of novel HSC-targeting ligands i.e. by transplantation assays, would definitely strengthen the overall impact of the manuscript. While we would welcome inclusion of such data, we think that these experiments are not mandatory.

If you feel you can satisfactorily deal with these points and those listed by the referees, you may wish to submit a revised version of your manuscript. Please attach a covering letter giving details of the way in which you have handled each of the points raised by the referees. A revised manuscript will be once again subject to review and you probably understand that we can give you no guarantee at this stage that the eventual outcome will be favorable.

Reviewer #1:

Summary

The authors have taken existing transcriptomics data on ligand and receptor expression of 12 different cell types in the hematopoietic system, and have used this to construct a cell-cell communication network, or map of what cell types are sending what signals to other receiving cells. The role of the different ligand and receptor genes is characterized based on existing gene ontology annotations. They divided their network into two parts: ligand production (senders) and ligand binding (receivers). Using a clustering of the ligand production data, they came to the conclusion that different cell types express ligands biased based on biological function (modular pattern). Using clustering of the ligand binding data, they find less modular interaction. The authors also address the factors of cell frequency and compartmentalization. Lastly, they performed a screen for 33 HSCe targeting ligands and determined their effects on cell fate.

General remarks

The authors do a good job with the statistical rigor of their computational analysis. For example, determining a false discovery rate for their definition of gene over-expression is a measure that is not always performed, but a nice precaution to take! The authors also do a nice job of describing the limitations/boundaries of their work. For example, they discuss that they do not have interaction partners for all ligands and receptors.

- Are you convinced of the key conclusions? I am convinced of the main conclusions of this paper. I think that the methods were well designed, implemented, and explained, and the results justified. However, I question the impact of the results found. I am not sure that any of the findings went beyond what was already assumed in the field. This seems to be a formal rigorous demonstration of what was already either demonstrated or believed to be true.

- What is the nature of the advance (conceptual, technical, clinical)? Technical - The biggest advance is the connectivity map that can be used broadly in the field when designing experiments to elucidate the structure of the signaling network.

- What audience will be interested in this study? The computational method here is not novel, so this paper strives to provide something new to the biological audience-those studying the hematopoietic system. The idea here is a good one: existing omics data is very difficult to parse through and conceptually understand. The authors strive to computationally/experimentally make sense of this data, and represent it in a way that would be much more intuitive to their audience. I am not convinced that they have done this. It is still difficult to draw biological insight from their figures. I think that their connectivity map can be presented in a more intuitive way to actually serve this need for their target audience.

Overall, I think that this is a strong body of work, but I think that the focus of the paper needs to be diverted from the small analytical conclusions made to the presentation of the cell-cell interaction map. The presentation and functional annotation of the map needs to be improved upon.

Major points

1. The authors make the conclusion that there is a modular pattern in the ligand production map. That is, different cell types produce ligands that fall in different functional categories. This conclusion does not represent a "new" idea. It is generally accepted that different cell types have different functions. I think this can only be made an interesting point if the authors refine their categories of "biological function" to be more narrow and specific - See Major point 2. If you look at Figure 3C, you see that though different categories of cell types have different expression patterns, you have not really come up with much insight. Almost all of the categories display almost all of the functions just at slightly varying levels. Right now the 12 cell types are lumped into 4 not-so-different groups, how do all 12 groups differ?

2. The use of "biological function" in this paper is very broad. They use categories such as apoptosis, cell cycle arrest, proliferation, etc. I think it is of no surprise or practical use to readers that different cell types function in these different categories. A more convincing result would

indicate the sub-processes, specific pathways, specific conditions, and specific temporal dynamics that these cell types are acting in. I think it is quite a stretch that the authors have "provided insight into the design principles of the hematopoietic system" given that all of their results are at such a high level. Perhaps they could bridge their results with more than just basic gene ontology. They should perform a literature search for studies done on different edges of their map and provide higher resolution on the "biological function" in their map.

3. The authors make the conclusion that the binding network displays less modularity. I am not convinced of this result. I think that without temporal or condition-specific expression data they are not necessarily finding the correct representation of receptor expression. If there is any temporal or condition specific data available, even a smaller data set, the authors should perform their analysis on that to justify obtaining the same result. Further, the lack of modularity really lies at the level of HSCe, the least differentiated cell. Perhaps this one cell type is ubiquitous in receptor expression, given that it does not have a defined function yet. The authors ought to remove the HSCe data from their analysis, and determine whether the remaining cell types display modularity.

4. This comment is in line with the above comments about the novelty of the author's claims. They do a nice job of exploring different explanations of their results. For instance, their use of cell frequency and compartmentalization to further explore HSCe regulation is convincing-although still not unexpected. Did anyone ever question that a higher abundance of cells would enable out competing of other cells for a given ligand? Did anyone ever question that a cell has to be physically close to a sender cell to receive its ligand? Seems a little obvious.

5. Overall, I think the strength of this paper lies not in the little analytical conclusions but in the construction of the cell-cell interaction map and the HSCe ligand screen. The most useful figure is Figure 6C, and an expansion of this figure to serve as a map that others can use to quickly look up different ligand-receptor interactions in this network would be most useful. Overlaying this map with a more in-depth characterization of biological function associated with the different signaling interactions will make this a truly impactful paper.

Minor points

1. The authors mention in their introduction that some ligands bind to receptor hetero-complexes while others act on receptors independently. The assumption is made in this paper that all interactions are of the independent nature. Ligand binding to a dimer, trimer, etc. of different receptors is not a rare event in mammalian signaling. The authors should mention the number of known hetero-receptor complexes in the hematopoietic system. Such complexes will affect their cell-cell connectivity map because if a receiving cell expresses only 1 or subset of the receptors required for a given ligand, a connection does not exist. If there are few such cases in this system, the authors should use this to justify their assumption. If there are many such cases in this system, the authors should factor this into their analytical methods.

2. Overall, I think that the paper is written well. The language is clear and easy to understand.

3. Figures such as Figure 2Ei provide no information at all. This network is the bread and butter of this paper and needs to be more informatively represented.

4. On page 9, the sentence: "...than in the ligand production network (Figure 2A)(t-test P-value..." should say Figure 3A.

5. On page 18, the sentence: "...and that technologies what target niche composition..." Should say "that target niche composition"

Reviewer #2:

Qiao et al. described a detailed directional cell-cell communication network of human hematopoietic cells. The network was based on previously published transcriptomic data of flow cytometrically

sorted distinct human hematopoietic cell populations. This study focused on ligand and receptor genes. By analyzing the network's property, the authors found an interesting organization principle: growth factor production is modular and associated with different cell types while receptor expression (hence growth factor binding) is promiscuous. The authors proposed other mechanisms (such as cell frequency and compartmentalization) were needed to achieve specific HSC fate. These mechanisms were probed by simulation. The simulation made several predictions that were experimentally confirmed previously (e.g. roles of Mono and Mega on HSC regulation). An *in vitro* screen was performed to test the roles of many candidate ligands on cultured HSC-e. Provisional feedback networks were constructed for cell type associated HSC fate regulation. Overall, the study was comprehensively done and is interesting. This study is an advancement (by using more defined cell population as resource of transcriptomic data) of the paper published from the same group (Kirouac et al MSB 2010). However, several concerns need to be addressed.

1. Many predictions were performed based the constructed networks. However, a large portion of the predictions are already experimentally proved/published previously. There are several novel predictions. It will show the utility of the method if the authors can make novel predictions and then validate using functional experiments (e.g. roles of TNFSF10 and TNF).
2. The *in vitro* screen of candidate ligands on HSC-e was informative. But the assay was merely phenotypic analyses of cell frequencies. It is well known that surface markers change during culture. So change of phenotypic cell population may not reflect actual change of the cells interested. The gold standard is to perform a transplantation assay. At least, the markers after culture used should be validated by transplantation. Or after the initial screen, a few of the prominent candidates should be validated using transplantation assay.
3. The authors made the assumption that all factors will directly regulate HSC-e. It could be the case that these factors directly regulate a third cell population and then indirectly regulate HSC-e. Can the authors distinguish these possibilities? The authors should acknowledge this in the text.
4. The transcriptomic data used to construct the networks were from freshly sorted cells from published papers. Yet, the *in vitro* experimental screen was performed on 'cultured' cells. There are significant differences between these two systems. Thus, it would make the networks more relevant if the authors used transcriptomic data from cultured cells as in their previous paper (Kirouac et al MSB 2010).
5. This study only looked at the intercellular regulation among hematopoietic cells, leaving out microenvironmental contribution. While it is reasonable to just study intercellular regulation of hematopoietic cells (in *in vitro* culture settings), the authors should tune down conclusions (in title and abstract particularly) to acknowledge the important contributions of the stromal compartment.

Reviewer #3:

Qiao and colleagues combined known ligand-receptor interactions with existing transcriptomic data from a variety of haematopoietic cell types at different stages of differentiation. From these data they constructed a network of ligand production and binding between the cell types, analysed its structure and attempted to validate some of their conclusions using a screening assay. They found that the ligand production network exhibits modularity with cell type-specific ligand production whilst the ligand binding network is more promiscuous. They then performed *in silico* simulations and the *in vitro* screen and concluded that additional restriction of the signals available to HSCs is required to confer specificity on feedback-mediated fate regulation.

The use of network analysis to study cell-cell communication is potentially useful and valuable in considering the control of fate decisions in situations such as the haematopoietic system. However, I am concerned that the initial construction of the network presented in this manuscript did not use an optimal strategy for determining which ligands and receptors are expressed by a particular cell type. In addition, I do not feel that the work addresses sufficiently the subtleties involved in whether a

particular cell is actually able to respond to a given ligand.

Furthermore, the *in vitro* screen does not, as it stands, fully convince me as a validation of the network.

Major points

Problems with construction of the network

In choosing the receptor and ligand genes that are expressed in each cell type, why was differential overexpression used rather than absolute expression? If a receptor/ligand is expressed then it is available to participate in the signalling network regardless of whether it is expressed in the other cell types chosen for comparison. The relative overexpression approach would risk ignoring receptors/ligands that are expressed in all (or the majority) of the cell types. It would be informative to know how many receptors/ligands such as these are left out from the network. Expressed genes could be absolutely identified using something like the HE/LE model in Hebenstreit et al. (2011). *Molecular Systems Biology*, 7, 497.

Why is 10% FDR "optimal" for classifying the over-expressed ligands and receptors? Inspection of supplementary figure 1 shows that this point (along with all the others) is very close to the line of no discrimination on the receiver operating characteristic curve. A classifier that produces results on this line is only as good as one that randomly assigns a class. As such, I would have concerns that the approach to identifying important ligands/receptors upon which the rest of the paper relies is inaccurate. A discussion of this by the authors would be valuable.

Finally, ignoring the role of hetero-multimeric receptors undermines the construction of the network and the conclusions that are drawn. If a cell does not express both components of a heterodimeric receptor, can it be said to respond to that receptor's cognate ligand? I have not performed an exhaustive survey of all the receptor/ligand combinations presented but, as an example, neutrophils and eosinophils are described in supplementary table 4 as responding to IL-7. This is due to their expression of IL2RG, the common gamma chain that forms the IL-7 receptor along with IL7R. IL7R is not expressed in these cells according to the data presented here and so I would argue that these cells are not responsive to IL-7. If I am interpreting the text correctly (pg 7), the approach taken by the authors would treat expression of IL7R and IL2RG as additive for response to IL7 but this makes no sense. Without both, the cell will not respond at all. In addition, neutrophils are described as responding to IL-2 due to their expression of IL2RG but they do not appear to express IL2RA or IL2RB.

As a final example, eosinophils are listed as responding to IL21 (again because of IL2RG) but they do not express IL21R which would be required for this to be the case.

The influence of cases such as these would be to increase the apparent promiscuity of the ligand binding network which is exactly what the authors observe.

I would like to see the authors make a greater effort to consider the role of hetero-multimeric receptors in the network and to adjust the model to take them into account such that a cell is only said to respond to a ligand if all of the required receptor components are present. This will be important if this work really is to provide a "...framework to systematically depict cell-cell communication as a network".

Figure 4F

As I interpret these graphs (and I might be wrong, they're not clear), each line represents the percentage of possible ligands with a particular function that are available to HSC-e cells given a varying probability that the ligands produced by a particular cell type are available to HSC-es. If this is the case, the value at $x=0$ is the number of ligands available if the particular cell type is not contributing anything. So, if it is high at $x=0$ and the line remains relatively flat then that cell type doesn't contribute many ligands no matter how "close" it is to the HSCe compartment. This appears to be the case for megakaryocyte apoptotic ligands (green line, top left graph); with zero probability of megakaryocytes contributing to the ligands, there are still ca. 66% of the set available to the HSC-es. This only increases to ca. 70% when $PMCN\text{-mega}=1$. It is hard to reconcile this with the

statement in the text that says "Our simulation results revealed the importance of Mega ligands to HSC-e apoptosis".

High-content screen

Text on page 14 states "At a working concentration of 100 ng/ml however the ligand led to a significant decrease in the number of HSC-enriched cells (P-value = 0.0007), progenitor cells (P-value = 0.0094) and mature cells (P-value = 0.0207) (Supplementary Figure 5B-ii) so it inhibited HSC-e proliferation, which is consistent with its well-known pro- apoptotic effect." Inhibition of proliferation is very different from promotion of apoptosis. Was apoptosis observed after treatment with this ligand?

In these data, TNFSF10 promotes self-renewal/proliferation at 10 ng/ml but is inhibitory (or maybe pro-apoptotic) at 100 ng/ml. In the Zamai (2000) paper cited, TNFSF10 causes measurable apoptosis of HL-60 cells at 10 ng/ml. Since HL-60 cells are mostly CD34- (Kuranda, K et al. 2011, J. Cell. Biochem., 112: 1277-1285) one might expect non-transformed mature cells to behave in a similar way and to exhibit a reduction in numbers.

Do the authors think that these discrepancies are a reflection of differences between cell types or does it represent a lack of sensitivity in their assay?

Data in figure 5C and supplementary table are presented as P values. To aid full interpretation of these results it would be helpful for the authors to show the magnitude of the change as well as its significance.

The selection criteria for ligands for the high-content screen includes using "literature anticipated HSC-targeted ligands". Nearly half of the selected ligands (15/33) have a literature reference in supplementary table 5. In some cases it seems that the results presented here simply recapitulate the existing data and confirm that these ligands do, indeed, have an effect upon HSCs. It is maybe, therefore, not surprising that there is a "significant enrichment of prediction capacity in this analysis". I would like to see the authors address this by commenting on i) whether their observations agree with published literature. ii) What the enrichment of prediction capacity is if the already-published ligands are ignored. iii) Whether the published ligands would have been chosen by the other two criteria dependent solely upon the ligand-receptor pairs and transcriptomic data. This would strengthen the authors' claims that the high-throughput screen was truly a validation of the network.

Discussion, pg17

It is unclear which data support the claim that "...the mature cells, particularly mono and granulocytes (neut, baso and eos) were found to mainly express inductive ligands for HSC-e proliferation and differentiation." Figure 6B indicates that monocytes are enriched for production of self-renewal inducing and proliferation inhibiting ligands; neutrophils for the same as well as quiescence inducing; eosinophils are only enriched for proliferation inhibiting ligands.

Minor points

Figure 2B

Shading for leukocyte chemotaxis gene expression in monocytes indicates a Z-score > 1.15 but no Z-score is given for this in supplementary table 1. Likewise, no Z-score is present in supplementary table 1 for PREB differentiation genes in PreB cells.

Generally, comparison between figure 2B and supplementary table 1 is impeded by the inconsistent ordering of cell types and gene types between the two tables.

Figure 6Ciii

The weighted network here is based on cell numbers for the mono-nucleated cell compartment. However, in figure 4Ci, the authors assert that due to cell numbers in this compartment, HSCs have a "negligible probability of accessing ligand resources in the system." Please could the authors explain the purpose of illustrating their network with this situation. Is it to emphasise the role of

spatial compartmentalisation? If so, this could be more clearly discussed.

Supplementary figure 2 legend

"Using the height of the primitive cell module...". Assume means "using the height of the..."

Supplementary figure 4 legend

"...setting the reference for cauterising the effects..." surely should be "...characterising the effects..."

Supplementary figure 5b

I found the use of circles of different sizes to represent cell numbers vs control hard to interpret. What is scaled relative to cell number? Radius? Area? Circumference? This figure can be easily interpreted as a two-dimensional measure (area) changing to represent a one-dimensional quantity (cell number). What's wrong with a bar chart to represent this?

Supplementary table 2

Poorly formatted. The final column is split onto a separate page.

Supplementary table 3

Receptors overexpressed in eosinophils are not listed

Supplementary table 7

What are the measurements of cell numbers? Is it $\times 10^3$?

Supplementary table 8

Poorly formatted. The final column is split onto a separate page.

Page 6

"Thus, we suspected the existence of gradients of receptor and ligand expression that paralleled the hematopoietic developmental hierarchy (Figure 2A)." Gradient is not a very helpful word here. When talking about cell-cell signalling I think of spatial concentration gradients. I think the authors mean "changes in the receptors/ligands expressed during progression through differentiation". A different word would help.

Page 8

"...supported the existence of 3 ligand-cell modules". Four modules are then listed.

Page 9

"...relative cell frequency to allow more abundant cell types skew the ligand resources...". I think there's a word missing/extra/out of order here.

Page 15

Several typographic errors. Please correct them.

Reviewer 1:

Summary

The authors have taken existing transcriptomics data on ligand and receptor expression of 12 different cell types in the hematopoietic system, and have used this to construct a cell-cell communication network, or map of what cell types are sending what signals to other receiving cells. The role of the different ligand and receptor genes is characterized based on existing gene ontology annotations. They divided their network into two parts: ligand production (senders) and ligand binding (receivers). Using a clustering of the ligand production data, they came to the conclusion that different cell types express ligands biased based on biological function (modular pattern). Using clustering of the ligand binding data, they find less modular interaction. The authors also address the factors of cell frequency and compartmentalization. Lastly, they performed a screen for 33 HSCe targeting ligands and determined their effects on cell fate.

General remarks

1. The authors do a good job with the statistical rigor of their computational analysis. For example, determining a false discovery rate for their definition of gene over-expression is a measure that is not always performed, but a nice precaution to take! The authors also do a nice job of describing the limitations/boundaries of their work. For example, they discuss that they do not have interaction partners for all ligands and receptors.

Thank you.

2. I am convinced of the main conclusions of this paper. I think that the methods were well designed, implemented, and explained, and the results justified.

Thank you.

3. However, I question the impact of the results found. I am not sure that any of the findings went beyond what was already assumed in the field. This seems to be a formal rigorous demonstration of what was already either demonstrated or believed to be true.

We do not agree with this reviewer's comment about the potential impact of our results. The goal of our study was to extract design principles of the cell-cell communication networks in the normal hematopoietic system and to gain in-depth understanding of feedback regulation of HSC fate. Learning the design principles of a heterogeneous system is important as demonstrated in microbiology where the rules encoded in molecular interaction are shown to stabilize cooperation among cells (Mol Syst Biol.

2011, 12;7:483).

Our main findings from the network analysis are that (i) hematopoietic cells produce ligands in a cell type-dependent manner, whereas ligand binding by hematopoietic cells is promiscuous; (ii) cell frequency and compartmentalization establish specificity in HSC fate regulation; (iii) HSC fate modulating signals are cell type associable; (iv) pathway analysis identified intracellular regulatory nodes enriched in cell type and ligand coupled responses. The findings summarize a design principle of the hematopoietic system: “Hematopoietic cells influence HSC towards certain cell fates by regulating the key intracellular pathway nodes through cell type-specific feedback signals. Further, control parameters such as relative cell frequency and local compartmentalization (niches) are opportunities to impose specificity in HSC fate regulation” (page 20 line 21 – page 21 line 2). Potential implications of this design principle for *in vitro* and *in vivo* hematopoiesis are also discussed (page 19 lines 7 – 21).

4. What is the nature of the advance (conceptual, technical, clinical)?
Technical - The biggest advance is the connectivity map that can be used broadly in the field when designing experiments to elucidate the structure of the signaling network.

We thank the reviewer for recognizing the technical advance of our work. In addition, we want to emphasize what we feel is a significant conceptual advance. Cell-cell communication is fundamental to the development, homeostasis, and disease initiation and progression in many tissues. Modeling cell-cell communication networks and learning new knowledge about biological processes from the models is a logical but challenging progression because of the large number of variables that need to be taken into account; see Youk and Lim, *Science*. 2014, 343(6171):1242782 for an example of a relatively simple cell-cell communication network. Focusing on HSC feedback regulation, we hypothesized that the design principle of hematopoietic cell-cell communication is important for our understanding of how HSC cell fate is regulated through feedback signaling. This led to the clustering analyses for identifying the structures of the ligand production network and the ligand binding network, and then the unexpected promiscuity in the ligand binding network led us to explore other mechanisms (e.g., cell frequency and spatial compartmentalization) to achieve specificity in HSC fate regulation. We anticipate that the concept is applicable to studies of other multi-cellular systems.

5. What audience will be interested in this study? The computational method here is not novel, so this paper strives to provide something new to the biological audience-those studying the hematopoietic system. The idea here is a good one: existing omics data is very difficult to parse through and conceptually understand. The authors strive to computationally /

experimentally make sense of this data, and represent it in a way that would be much more intuitive to their audience.

I am not convinced that they have done this. It is still difficult to draw biological insight from their figures. I think that their connectivity map can be presented in a more intuitive way to actually serve this need for their target audience.

Thank you for the comment, which we worked hard to address. Instead of presenting the cell-cell communication network as a “hairball” of cell-ligand-cell connections, we now show the construction and analysis of the network in a more intuitive manner. See Figure 2D for the revision.

In addition, we included the pathway information of the ligands into the map of feedback regulation of HSC fate (Figure 7). Specifically, we performed a pathway enrichment analysis for the ligands of different directive roles towards HSC cell fate (Figure 7A) using data from a manually curated database (<http://thomsonreuters.com/metacore/>). Results of the analysis pointed to ligand function-associated intracellular regulatory motifs. Specifically, ligands that drove HSC quiescence, self-renewal, proliferation-induction and proliferation-inhibition were enriched in SMAD, NF- κ B, STAT and caspase cascade pathways, respectively (Figure E9, Figure 7B). This new layer of information warrants future studies towards understanding how exogenous ligands direct HSC fate decisions. See information about the identification of these intracellular regulatory motifs (page 16 line 18 – page 17 line 9) and the impact of the findings (page 20 lines 1 – 17) in the revised manuscript.

6. Overall, I think that this is a strong body of work, but I think that the focus of the paper needs to be diverted from the small analytical conclusions made to the presentation of the cell-cell interaction map. The presentation and functional annotation of the map needs to be improved upon.

Thank you. We have added text to the manuscript to emphasize the novel contributions of our study; see our response to this reviewer’s general remark 3 for details.

Major points

1. The authors make the conclusion that there is a modular pattern in the ligand production map. That is, different cell types produce ligands that fall in different functional categories. This conclusion does not represent a “new” idea. It is generally accepted that different cell types have different functions. I think this can only be made an interesting point if the authors refine their categories of “biological function” to be more narrow and specific - See Major point 2. If you look at Figure 3C, you see that though different categories of cell types have different expression patterns, you

have not really come up with much insight. Almost all of the categories display almost all of the functions just at slightly varying levels. Right now the 12 cell types are lumped into 4 not-so-different groups, how do all 12 groups differ?

The reviewer commented on the novelty of our conclusion that "*hematopoietic cells produce ligands in a cell type-dependent manner*". While it is accepted that different blood cell types perform different functions, to the best of our knowledge, the phenomenon has not been systemically associated with ligand production of individual cell types. The conclusion is important because it is a piece to the puzzle of the design principle of the normal hematopoietic cell-cell communication network, understanding which is the goal of our study (particularly with respect to feedback regulation of HSC fate). See our response to this reviewer's general remark 3 for details.

The reviewer suggested to "*refine the categories of biological function to be more narrow and specific*". We agree with the suggestion. We have redefined our categories of biological functions. The new categorization provided more insight into the ligands produced by each cell type than our previous version. See our response to this reviewer's major point 2 for details.

2. The use of "biological function" in this paper is very broad. They use categories such as apoptosis, cell cycle arrest, proliferation, etc. I think it is of no surprise or practical use to readers that different cell types function in these different categories. A more convincing result would indicate the sub-processes, specific pathways, specific conditions, and specific temporal dynamics that these cell types are acting in. I think it is quite a stretch that the authors have "provided insight into the design principles of the hematopoietic system" given that all of their results are at such a high level. Perhaps they could bridge their results with more than just basic gene ontology. They should perform a literature search for studies done on different edges of their map and provide higher resolution on the "biological function" in their map.

Thank you for this comment, which we have worked hard to address. We have broken down our biological function into the following categories to increase the resolution of our ligand function analysis:

(a) Survival related functions

- Survival inhibition by external signals
- Survival mediated by external signals via PI3K-AKT, MAPK and JAK-STAT, and NF- κ B

(b) Cell migration related functions

- Leukocyte chemotaxis

- Regulation of angiogenesis
- (c) Immune related functions
 - Innate inflammatory response
 - T helper cell differentiation
- (d) Cell cycle regulation
 - G1-S regulation
 - G2-M regulation

Functional enrichment results for the ligands produced by each cell type are shown in Figure 3C. The following sentences have been included in the revised manuscript (page 9 lines 1 – 8), “For instance, ligands of the neutrophil-monocyte module enriched in exogeneous signals that inhibit cell survival (HG Z-scores were 1.63 and 2.98 for Mono and Neut, respectively), and signals that mediate cell survival via NF-κB (HG Z-scores were 2.15 and 1.43 for Mono and Neut, respectively); ligands of Baso, Eos and PreB within the (Boso + Eos + Mega + PreB) module enriched in signals that direct differentiation cell fates of T helper cells (HG Z-scores were 1.17, 2.65 and 3.18 for Baso, Eos and PreB, respectively); and ligands of EryB enriched in signals that regulate G1-S cell cycle transition (HG Z-score = 1.41) (Figure 3C). See Table E6 for the other HG enrichment Z-scores”.

It is noteworthy that the biological function-associated gene sets were obtained from a manually curated pathway database (MetaCore: <http://thomsonreuters.com/metacore/>). Despite the high quality of the database, relevance of the ligands to human HSC required experimental validation. We thus performed a comprehensive *in vitro* study for the ligands of interest on the HSC-enriched (HSC-e, Lin⁻CD34⁺CD38⁻Rho^{low}CD45RA⁻CD49f⁺) cells. Results of the *in vitro* study allowed us to categorize HSCe-targeting ligands into functional categories such as HSC-e quiescence induction, self-renewal induction, differentiation induction, proliferation induction, and proliferation inhibition. Pathway enrichment analysis was subsequently performed for ligands of each functional category. Our results suggest the existence of ligand function-associated intracellular regulatory motifs. Specifically, the HSC-e quiescence-inducing ligands were enriched in SMAD pathway; the self-renewal-inducing ligands were enriched in NF-κB pathway; the proliferation-inducing ligands were enriched in STAT pathway; and the proliferation-inhibition ligands were enriched in caspase cascades (Figure E9, Figure 7B). See information about the identification of these intracellular regulatory motifs (page 16 line 18 – page 17 line 9) and the impact of the findings (page 20 lines 1 – 17) in the revised manuscript.

3. The authors make the conclusion that the binding network displays less modularity. I am not convinced of this result. I think that without temporal or condition-specific expression data they are not necessarily finding the

correct representation of receptor expression. If there is any temporal or condition specific data available, even a smaller data set, the authors should perform their analysis on that to justify obtaining the same result.

We interpret the reviewer's comment as that the reviewer is concerned with the robustness of the structure of the ligand binding network. To address the reviewer's comment, we compared the receptor gene expression of cord blood-derived Mega and that of culture-derived Mega. In our recent study published in PLoS Comp Biol.2012;8(2):e1002838, we profiled gene expression of Mega isolated from human cord blood samples and Mega isolated from a 4-day cell culture supplemented with the same basal cytokines as we used in the current *in vitro* experiments. We found that receptor gene expression profiles of the two Mega samples were correlated ($R^2 = 0.73$), although it is noteworthy that expression of ~20% of the differentially over-expressed receptor genes in the cord blood-derived Mega were changed by at least 2-fold (mainly associate with metabolism) (Figure R1.1). To this end, we have emphasized throughout the manuscript that our network structure was identified from the gene expression data of blood cell samples isolated from uncultured human umbilical cord blood. Furthermore, we have included a sentence in the Discussion to highlight the need to understand time-dependent and / or condition-dependent network structures, "A final limitation of this study is that we used a static (human UCB) network to predict potentially dynamic feedback relationships between HSC-e and the other cell types. Exploring how the network connections change during culture evolution (Qiao *et al*, 2012) is an important next step" (page 22 lines 1 – 4).

Figure R1.1. Correlation analysis for the receptor expression of human umbilical cord blood-derived megakaryocytes (Mega, CD34⁺CD41⁺CD61⁺CD45⁺) and *in vitro* culture-derived Mega. Red: differentially over-expressed receptor genes (n = 35) in umbilical cord blood-derived Mega comparing to 11 other blood cell types analyzed in this study.

Further, the lack of modularity really lies at the level of HSCe, the least differentiated cell. Perhaps this one cell type is ubiquitous in receptor

expression, given that it does not have a defined function yet. The authors ought to remove the HSCe data from their analysis, and determine whether the remaining cell types display modularity.

We agree that our results may be specific to the HSCe; in fact, this was our goal as HSCe is the most difficult to analyze because of its rarity, and the cell type is the most important in the whole hematopoietic system because the system will collapse without sustaining the HSCe population. Therefore, we do not think that it is reasonable to remove HSCe from our clustering analysis. To clarify that promiscuity is a property of the ligand binding network of the hematopoietic system as a whole, the following sentence is included in revised manuscript, "... ubiquitously shared ligand binding amongst the 12 cell types..." (page 9 line 22 – page 10 line 1).

Despite our position that HSCe should not be excluded from the clustering analysis, we tried to cluster the ligand binding network without HSCe to address this reviewer's comment. Promiscuous ligand binding is evident from Figure R1.2.

Figure R1.2. Hierarchical clustering analysis for the ligand binding network (false discovery rate, FDR = 10%) without HSCe based on Euclidean distances.

4. This comment is in line with the above comments about the novelty of the author's claims. They do a nice job of exploring different explanations of their results. For instance, their use of cell frequency and compartmentalization to further explore HSCe regulation is convincing—although still not unexpected. Did anyone ever question that a higher abundance of cells would enable out competing of other cells for a given ligand? Did anyone ever question that a cell has to be physically close to a sender cell to receive its ligand? Seems a little obvious.

We are glad our results appear obvious, which in retrospect is often the case. Despite obviousness of the answers, focusing on the hematopoietic system, our contribution is to show how all the answers fit together to achieve coordinated regulation of HSC fate. This system level

understanding is important as shown by the example of the monocyte and granulocyte populations in our manuscript. According to our analysis, monocytes and granulocytes (including neutrophils, basophils and eosinophils) were the major cell populations in mono-nucleated cells isolated from human cord blood samples. These two cell populations expressed mainly inhibitory signals for HSC-e proliferation and inducing signals for HSC-e differentiation. Combination of these two types of signals can exhaust the HSC population. However, under a normal *in vivo* condition, monocytes and granulocytes mainly circulate in the peripheral tissues. Thus, their secreted factors have only limited access to HSC because of the blood-bone marrow barrier; see page 19 (line 7 – line 13) of the manuscript for details. Furthermore, the identified design principle shed insights into our existing systems for *in vitro* HSC expansion (Cell Stem Cell. 2012;10(2):218-229) (page 19 lines 13 – 21). From our prospective of view, further investigation into the design principle may lead us to answers of big questions; for example, “how defects in HSC niche composition and physical structure, or defects in HSC intracellular regulatory motifs impacts feedback regulation to HSC fate decisions *in vivo*, and consequently causes leukemogenesis” (page 21 lines 2 – 6).

5. Overall, I think the strength of this paper lies not in the little analytical conclusions but in the construction of the cell-cell interaction map and the HSCe ligand screen.

Thank you for recognizing the importance of our work.

The most useful figure is Figure 6C, and an expansion of this figure to serve as a map that others can use to quickly look up different ligand-receptor interactions in this network would be most useful. Overlaying this map with a more in-depth characterization of biological function associated with the different signaling interactions will make this a truly impactful paper.

We have interpreted the reviewer’s comment that “*in-depth characterization of biological function associated with the different signaling interactions*” in two ways.

The first interpretation asks to gain more understanding of how the ligands in Figure 7A (the original Figure 6C) act from a signaling pathway perspective. To address this point, we performed a pathway enrichment analysis for the ligands. The results pointed to ligand function-associated intracellular regulatory motifs. See our response to this reviewer’s general remark 5 for details.

The second interpretation points to the need to perform functional experiments to understand the mechanism of action of each ligand in the

feedback network (i.e., inducing HSC-e quiescence, self-renewal, differentiation or proliferation). While we agree that performing functional experiments for the ligands is a logical step, we want to point out the goal of the current study is to extract design principles of the cell-cell communication networks in the normal hematopoietic system and to gain in-depth understanding of feedback regulation of HSC fate (see our response to this reviewer's general remark 3). Taking the advantage of short duration and high throughput, we chose to use phenotype-based *in vitro* experiments to learn the effects of HSCe-targeting ligands in the network on HSC-e cell fate. We cultured HSC-e, the most enriched HSC population to our knowledge, for a short time (7 days) under supplementation of the ligands of interest. To quantify the effect the ligands on HSC-e self-renewal, we used the expression of CD34⁺CD133⁺CD90⁺, a marker set whose fidelity in predicting transplantable HSC post cell culture has been assessed (Blood. 2010; 115(2):257-60, Blood. 1994; 83:2410-7, Exp Hematol. 2001; 29:1465-73, Blood. 2000; 95: 102-110). To quantify the effects of the ligands on progenitor cell generation, we used the cell surface marker combination (CD34⁺133⁻ or CD90⁻), which has the ability to enumerate progenitors in culture-derived cells as we have validated using the functional colony-forming cell (CFC) assay (Figure E5). In spite of our efforts in using the well-studied cell surface marker combination for culture-derived transplantable HSC and the CFC-validated surface marker combination for progenitors, we concede that further *in vitro* and / or *in vivo* functional analysis of specific interesting and novel ligands is important (ideally using limiting dilution primary and secondary *in vivo* transplant assays as we have previously published on). To this end, we have included the following sentences in the revised manuscript (page 21 line 15 – page 22 line 1), "A second limitation of this study is that we defined the exogenous effects of 33 ligands on HSC-e fate decision according to *in vitro* measurements of a cell surface marker expression-based phenotypic assay ... Further functional validation of the surface markers to cell function fidelity would certainly strengthen our analysis of network directionality; ultimately, our network should guide the selection of potentially novel HSC-e regulating cell types, ligands, and their key intracellular signaling nodes for in-depth *in vivo* characterization".

Minor points

1. The authors mention in their introduction that some ligands bind to receptor hetero-complexes while others act on receptors independently. The assumption is made in this paper that all interactions are of the independent nature. Ligand binding to a dimer, trimer, etc. of different receptors is not a rare event in mammalian signaling. The authors should mention the number of known hetero-receptor complexes in the hematopoietic system. Such complexes will affect their cell-cell

connectivity map because if a receiving cell expresses only 1 or subset of the receptors required for a given ligand, a connection does not exist. If there are few such cases in this system, the authors should use this to justify their assumption. If there are many such cases in this system, the authors should factor this into their analytical methods.

To our knowledge, 15 class-1 cytokines, out of 253 ligands that we are interested in, require hetero-multimeric receptors to initiate intracellular signaling. The ligands and their hetero-multimeric receptors are the followings: CSF2 (CSF2RA-CSF2RB), IL3 (IL3RA-CSF2RB), IL5 (IL5RA-CSF2RB), IL6 (IL6R-IL6ST), IL11 (IL11RA-IL6ST), LIF (LIFR-IL6ST), OSM (OSMR-IL6ST), CNTF (CNTFR-LIFR-IL6ST), IL2 (IL2RB-IL2RG, IL2RA-IL2RB, IL2RA-IL2RB-IL2RG), IL4 (IL4R-IL2RG), IL21 (IL21R-IL2RG), IL13 (IL13RA1-IL4R), IL7 (IL7R-IL2RG), IL9 (IL9R-IL2RG), IL15 (IL15RA-IL2RB-IL2RG). We reconstructed the cell-cell communication network after implementing the condition that a connection between a ligand and a cell type only exists when all the receptors of a complex are differentially over-expressed in the cell type, and clustering was performed on the new ligand production network (Figure E2C) and the new ligand binding network (Figure E3C). The results recapitulated our early observation, i.e., cell-to-ligand interactions in the ligand production network are modular, and the ligand-to-cell interactions in the ligand binding network are promiscuous.

To summarize the above points, the following sentences are included in the revised manuscript, “Fifteen class-1 cytokines including CNTF, CSF2, IL2, IL3, IL4, IL5, IL6, IL7, IL9, IL11, IL13, IL15, IL21, LIF and OSM require interaction with hetero-multimeric receptors to initiate intracellular signaling cascades (Robb, 2007). Given that our network was constructed from gene expression data and it is possible that receptor proteins encoded by lowly expressed genes are functional for signal transduction (Schumann *et al*, 1996). From a modeling perspective, we assumed that the greater the number of receptor species that a cell expresses for a ligand, the higher the probability that the ligand binds to the cell. We considered the interactions of each ligand and its cognate receptors independently...” (page 7 lines 4 – 12).

Regarding the results of our network structure, the following sentences have been included in the manuscript:

- For the ligand production network, “...this conclusion is robust to the choice of FDR threshold for differential gene over-expression (Figure E2B) and the incorporation of hetero-multimeric receptors in network construction (Figure E2C)” (page 9 lines 11 – 13).
- For the ligand binding network, “...the promiscuous network structure is robust to the choice of FDR for differential gene over-expression (Figure E3B) and the incorporation of hetero-multimeric

receptors in network construction (Figure E3C)" (page 10 lines 2 – 4).

2. Overall, I think that the paper is written well. The language is clear and easy to understand.

Thank you.

3. Figures such as Figure 2Ei provide no information at all. This network is the bread and butter of this paper and needs to be more informatively represented.

To address this comment the old Figure 2 has been reorganized. See new Figure 2D as a replacement for the old Figure 2Ei.

4. On page 9, the sentence: "...than in the ligand production network (Figure 2A)(t-test P-value..." should say Figure 3A.

Corrected

5. On page 18, the sentence: "...and that technologies what target niche composition..." Should say "that target niche composition"

Corrected

Reviewer 2:

Qiao et al. described a detailed directional cell-cell communication network of human hematopoietic cells. The network was based on previously published transcriptomic data of flow cytometrically sorted distinct human hematopoietic cell populations. This study focused on ligand and receptor genes. By analyzing the network's property, the authors found an interesting organization principle: growth factor production is modular and associated with different cell types while receptor expression (hence growth factor binding) is promiscuous. The authors proposed other mechanisms (such as cell frequency and compartmentalization) were needed to achieve specific HSC fate. These mechanisms were probed by simulation. The simulation made several predictions that were experimentally confirmed previously (e.g. roles of Mono and Mega on HSC regulation). An *in vitro* screen was performed to test the roles of many candidate ligands on cultured HSC-e. Provisional feedback networks were constructed for cell type associated HSC fate regulation. Overall, the study was comprehensively done and is interesting. This study is an advancement (by using more defined cell population as resource of transcriptomic data) of the paper published from the same group (Kirouac et al MSB 2010).

Thank you for your supportive comments.

However, several concerns need to be addressed.

1. Many predictions were performed based the constructed networks. However, a large portion of the predictions are already experimentally proved/published previously. There are several novel predictions. It will show the utility of the method if the authors can make novel predictions and then validate using functional experiments (e.g. roles of TNFSF10 and TNF). Further, *in vitro* screen of candidate ligands on HSC-e was informative. But the assay was merely phenotypic analyses of cell frequencies. It is well known that surface markers change during culture. So change of phenotypic cell population may not reflect actual change of the cells interested. The gold standard is to perform a transplantation assay. At least, the markers after culture used should be validated by transplantation. Or after the initial screen, a few of the prominent candidates should be validated using transplantation assay.

Thank you for the comment. While we agree that performing functional experiments for prominent ligand candidates from the *in vitro* experiments is a logical step, we want to point out the goal of the current study is to extract design principles of the cell-cell communication networks in the normal (umbilical cord blood) hematopoietic system and to gain in-depth understanding of feedback regulation of HSC fate.

The reviewer also commented on the discrepancy between cell surface

marker expression and cell function in cultured cells. We attempted to address the comment by quantifying the cell surface marker usage for HSC-enriched cells and progenitor cells, respectively. First, regarding the use of CD34⁺CD133⁺CD90⁺ for quantifying HSC-enriched cells. Numerous transplantation studies have been performed to assess the fidelity of CD34, CD133 and CD90 expression in cultured cells. For instance, our group has shown that CD133⁺ marker identifies culture-derived SRC and LTC-IC in human cord blood Lin⁻ cell-initiated culture (Blood. 2010; 115(2):257-60). Other groups have also shown that the expression of CD90⁺CD34⁺ is one of the most reliable predictors for culture-derived transplantable HSC (Blood. 1994; 83:2410-7, Exp Hematol. 2001; 29:1465-73, Blood. 2000; 95: 102-110). Therefore, we believe that CD34⁺CD133⁺CD90⁺ expression is a suitable surrogate marker to track candidate HSC during culture. The above-mentioned references have been included in the revised manuscript (page 13 lines 14 – 15). Second, we defined progenitor cells by the cell surface marker combination (CD34⁺133⁻ or CD90⁻). Using IL11 and CSF2 as representative ligands, we quantified the outputs of this marker combination using the functional colony-forming cell (CFC) assay for progenitor cell quantification. As shown in Figure R2.1, the relative progenitor cell outputs measured by the surface marker combination correlated with the outputs from the CFC assay. This result provided confidence in our use of (CD34⁺133⁻ or CD90⁻) cells to quantify progenitor cell outputs of our cell culture. The result is included in Figure E5 of the revised manuscript.

To sum up, in spite of our efforts in using well-studied cell surface marker combinations for culture-derived transplantable HSC and the CFC-validated surface marker combination for progenitors, we concede that further *in vitro* and / or *in vivo* functional analysis of specific interesting and novel ligands is important (ideally using limiting dilution primary and secondary *in vivo* transplant assays as we have previously published on). To this end, we have included the following sentences in the revised Discussion, “A second limitation of this study is that we defined the exogenous effects of 33 ligands on HSC-e fate decision according to *in vitro* measurements of a cell surface marker expression-based phenotypic assay ... Further functional validation of the surface markers to cell function fidelity would certainly strengthen our analysis of network directionality; ultimately, our network should guide the selection of potentially novel HSC-e regulating cell types, ligands, and their key intracellular signaling nodes for in-depth *in vivo* characterization” (page 21 line 15 – page 22 line 1).

Figure R2.1. Comparison between progenitor cell counts obtained using a cell surface marker expression-based phenotypic assay and the functional colony-forming cell assay. Shown are mean \pm sd ($n = 33$ for the phenotypic data of CSF2; $n = 5$ for the phenotypic data of IL11; $n = 2$ for the functional data).

2. The authors made the assumption that all factors will directly regulate HSC-e. It could be the case that these factors directly regulate a third cell population and then indirectly regulate HSC-e. Can the authors distinguish these possibilities? The authors should acknowledge this in the text.

We argue that the tested factors mainly regulate HSC-e for two reasons. First, receptor gene expression levels of the tested ligands are higher in HSCe than in many of the other analyzed cell types. We have used the receptor gene expression level as one of the criteria to select test ligands as described in line 6 on page 13 and shown in Figure E4 in the revised manuscript. Second, our culture was only 7 days compared to at least 12 days in other *in vitro* HSC expansion studies (Cell Stem Cell. 2012;10(2):218-229). The short culture time would reduce feedback effect in our culture readouts (as we have shown also in Blood.2014;123(5):650-8). Nevertheless, we cannot rule out the possibility of indirect regulation of HSC-e in our culture. We have included the following sentences in the revised Discussion, “Although our *in vitro* study was specifically designed to enrich for direct effects of ligands on HSC-e by using the HSCe receptor expression information as a criterion for selecting test ligands and using a short culture time (7 days) (Csaszar *et al*, 2014), further analysis of multi-step and adaptive feedback is needed to strengthen links to *in vivo* hematopoiesis” (page 22 lines 5 – 9).

3. The transcriptomic data used to construct the networks were from freshly sorted cells from published papers. Yet, the *in vitro* experimental screen was performed on 'cultured' cells. There are significant differences between these two systems. Thus, it would make the networks more relevant if the authors used transcriptomic data from cultured cells as in their previous paper (Kirouac *et al* MSB 2010).

We apologize for the confusion associated with our experimental protocol.

In fact, all the analysis in this study was performed with cells isolated directly from cord blood samples and treated directly with cytokines. The input cell population is consistent with the gene expression data that we used for network construction. We have emphasized this in the manuscript, “We examined the phenotypic impact of each ligand on 40 HSC-enriched cells (HSC-e: human UCB Lin⁻CD34⁺Rho^{low}CD38⁻CD45RA⁻CD49f⁺) isolated from human UCB samples...” (page 13 lines 7 – 9).

4. This study only looked at the intercellular regulation among hematopoietic cells, leaving out microenvironmental contribution. While it is reasonable to just study intercellular regulation of hematopoietic cells (in in vitro culture settings), the authors should tune down conclusions (in title and abstract particularly) to acknowledge the important contributions of the stromal compartment.

We acknowledge the importance of microenvironmental contribution to HSC fate control. We think that our hematopoietic cell-based intercellular regulation of HSC fate is complementary to the bone marrow niche cell regulation. We have strengthened this in the Discussion, “Extensive effort has been made to understand how stromal cells in the bone marrow microenvironment regulate HSC fate decisions (Zhang *et al*, 2003; Nakamura *et al*, 2010; Kunisaki *et al*, 2013). In addition, we propose a hematopoietic cell-driven feedback system that regulates HSC fate decisions through intercellular signaling” (page 18 lines 8 – 11). Furthermore, we replaced the use of “hematopoietic system” with “umbilical cord blood” in the Title, and we clarified that our study is based on human umbilical cord blood hematopoiesis in the revised Abstract (page 2 line 6).

Reviewer 3:

Qiao and colleagues combined known ligand-receptor interactions with existing transcriptomic data from a variety of haematopoietic cell types at different stages of differentiation. From these data they constructed a network of ligand production and binding between the cell types, analysed its structure and attempted to validate some of their conclusions using a screening assay. They found that the ligand production network exhibits modularity with cell type-specific ligand production whilst the ligand binding network is more promiscuous. They then performed *in silico* simulations and the *in vitro* screen and concluded that additional restriction of the signals available to HSCs is required to confer specificity on feedback-mediated fate regulation.

1. The use of network analysis to study cell-cell communication is potentially useful and valuable in considering the control of fate decisions in situations such as the haematopoietic system. However, I am concerned that the initial construction of the network presented in this manuscript did not use an optimal strategy for determining which ligands and receptors are expressed by a particular cell type. In addition, I do not feel that the work addresses sufficiently the subtleties involved in whether a particular cell is actually able to respond to a given ligand.

We thank the reviewer for recognizing the value of our work.

The reviewer questioned the use of differentially over-expressed genes of each cell type for network construction. We have assumed that differentially over-expressed ligand and receptor genes of each cell type are more representative of the cell's identity and biological properties. We have listed our reasons and results of a new analysis in the response to this reviewer's major point 1.

2. Furthermore, the *in vitro* screen does not, as it stands, fully convince me as a validation of the network.

Our interpretation of the comment is based on the reviewer's major point 7. The reviewer underlined that "*the selection criteria for ligands for the high content screen include literature anticipated HSC-targeted ligands*", and therefore our *in vitro* experiments do not serve as validation for HSCe-targeting ligands identified from the gene expression-based network analysis. The reviewer has raised a valid concern. We apologize for the confusion associated with our description of the selection criteria for the tested ligands. Please see our response to this reviewer's major point 7 for details.

Major points

Problems with construction of the network

1. In choosing the receptor and ligand genes that are expressed in each cell type, why was differential overexpression used rather than absolute expression? If a receptor/ligand is expressed then it is available to participate in the signalling network regardless of whether it is expressed in the other cell types chosen for comparison. The relative overexpression approach would risk ignoring receptors/ligands that are expressed in all (or the majority) of the cell types. It would be informative to know how many receptors/ligands such as these are left out from the network. Expressed genes could be absolutely identified using something like the HE/LE model in Hebenstreit et al. (2011). *Molecular Systems Biology*, 7, 497.

Differential overexpression, as opposed to absolute expression, was used for two reasons. First, Figure 2d in *Nature* 2011;473(7347):377 indicates that there is very weak correlation between mRNA and protein expression levels for lowly expressed gene products, but the randomness tapers off as the mRNA level increases. We thus assumed that gene products with high mRNA expression levels display stronger correlation between RNA and protein expression levels. Second, we kept it consistent with the general assumption of molecular biology studies where the up- and down-regulated genes under normal and diseased conditions, or under treated and control conditions are performed. We assumed that differentially over-expressed ligands and receptors of each cell type are more representative of the cell's identity and biological properties than ubiquitously expressed ligands and receptors. We think that the assumptions are reasonable because the goal of our work is to identify and contrast the properties of the ligand producing network and that of the ligand binding network, only the differential receptors/ligands of each cell type will contribute to our conclusion. To summarize our points, we included the following sentences in the revised manuscript, "In the construction of CCC networks, we assumed that the differentially over-expressed genes of each cell type are predictive of the cell type's protein expression (Schwanhausser *et al*, 2011), and representative of the cell type's biological properties" (page 6 lines 13 – 15).

The use of differentially over-expressed ligands and receptors left out 59 ligands and 62 receptors in our ligand-receptor database from network construction because the ligands and receptors were not differentially over-expressed by any cell types of interest.

Despite our position that it is valid to use differential overexpressed ligand and receptor genes for network construction, we tried the HE/LE model on

our data. In this study, we had to combine Illumina (for hematopoietic stem and progenitor cells) and Affymetrix (for mature cells) datasets. The binding affinity between pre-designed probes and targeting genes varies among probes in oligo-microarrays, and thus it is incorrect to combine the absolute expression levels of the two datasets. To overcome the issue in probe binding affinities, we normalized data of each platform to the data of pro-B cells that were profiled by the two platforms. We did not observe bimodal distribution in the normalized data as shown in Figure R3.1. Thus, the distribution model is not applicable to our data.

Figure R3.1. Gene expression distribution of 12 blood cell types of interest on the log₁₀ scale. HSCe: HSC-enriched (Lin⁻CD34⁺CD38⁻CD45RA⁻CD49f⁺CD90^{+/+}) cells; CMP: common myeloid progenitors; MEP: megakaryocyte-erythroid progenitors; GMP: granulocyte-monocyte progenitors; EryB: erythroblasts; Mega: megakaryocytes; Neut: neutrophils; Baso: basophils; Eos: eosinophils; Mono: monocytes; MLP: multilymphoid progenitors; PreB: precursor B cells.

2. Why is 10% FDR "optimal" for classifying the over-expressed ligands and receptors? Inspection of Figure E1 shows that this point (along with all the others) is very close to the line of no discrimination on the receiver operating characteristic curve. A classifier that produces results on this line is only as good as one that randomly assigns a class. As such, I would have concerns that the approach to identifying important ligands/receptors upon which the rest of the paper relies is inaccurate. A discussion of this by the authors would be valuable.

When constructing the network, we were concerned about missing potentially interesting ligands and receptor by using a too stringent threshold (high true positive rate and low false positive rate) for a differential over-expression analysis. At the same time, we wanted to minimize noise in the network. Therefore, we chose the FDR threshold that gave "equal" true positive rate and false positive rate. The FDR of 10% seems to give us a good representation of ligand and receptor gene expression. For instance, FLT3LG is known to be a growth factor that affects multipotential hematopoietic cells. Expression of the receptor on common myeloid progenitors (CMP) was not detected at FDR 5% but detected at FDR 10%. To clarify the use of 10% FDR, we included the following sentences in the manuscript, "To determine an appropriate false discovery rate (FDR) to define differential over-expression, we tested FDRs of 1%, 5%, 10%, 20% and 25%, and then compared the set of receptors identified at each threshold to a benchmark of known cell type-associated receptors (see Materials and Methods). A FDR of 10% detected the known cell type-associated receptors with the optimal combination of sensitivity and specificity (Figure E1), and thus the ligands (Table E3A) and receptors (Table E3B) differentially over-expressed according to this threshold were used in the subsequent analyses (Figure 1; step 1b)" (page 6 lines 15 – 21).

To be confident with the conclusions of our network structures, we have also constructed and analyzed networks for ligands and receptors obtained at FDR of 1%, 5%, 15%, 20% and 25%. The results are shown in Figure E2B and Figure E3B in the initial submission. Our conclusions regarding the network structure are not affected by the FDR values as we mentioned in the manuscript:

- For the network of ligand production: "This conclusion is robust to the choice of FDR threshold for differential gene over-expression (Figure E2B)..." (page 9 lines 11 – 12).
- For the network of ligand binding: "The promiscuous network structure is robust to the choice of FDR for differential gene over-expression (Figure E3B)..." (page 10 lines 2 – 3).

3. Finally, ignoring the role of hetero-multimeric receptors undermines the construction of the network and the conclusions that are drawn. If a cell does not express both components of a heterodimeric receptor, can it be said to respond to that receptor's cognate ligand? I would like to see the authors make a greater effort to consider the role of hetero-multimeric receptors in the network and to adjust the model to take them into account such that a cell is only said to respond to a ligand if all of the required receptor components are present. This will be important if this work really is to provide a "...framework to systematically depict cell-cell communication as a network".

To our knowledge, 15 class-1 cytokines, out of 253 ligands that we are interested in, require hetero-multimeric receptors to induce intracellular signaling cascade (page 7 lines 4 – 7 in the manuscript). We agree that a cell will not respond to these cytokines without expression of all the receptor components. We reconstructed the cell-cell communication network after implementing the condition that a connection between a ligand and a cell type only exists when all the receptors of a complex are differentially over-expressed in the cell type, and clustering was performed on the new ligand production network (Figure E2C) and the new ligand binding network (Figure E3C). The results recapitulated our early observation, i.e., cell-to-ligand interactions in the ligand production network are modular, and the ligand-to-cell interactions in the ligand binding network are promiscuous.

To summarize the new results, we included the following sentences in the revised manuscript,

- For the ligand production network, "...this conclusion is robust to the choice of FDR threshold for differential gene over-expression (Figure E2B) and the incorporation of hetero-multimeric receptors in network construction (Figure E2C)" (page 9 lines 11 – 13).
- For the ligand binding network, "...the promiscuous network structure is robust to the choice of FDR for differential gene over-expression (Figure E3B) and the incorporation of hetero-multimeric receptors in network construction (Figure E3C)" (page 10 lines 2 – 4).

I have not performed an exhaustive survey of all the receptor/ligand combinations presented but, as an example, neutrophils and eosinophils are described in Table E4 as responding to IL-7. This is due to their expression of IL2RG, the common gamma chain that forms the IL-7 receptor along with IL7R. IL7R is not expressed in these cells according to the data presented here and so I would argue that these cells are not responsive to IL-7.

We treated the interactions between a ligand and its cognate receptors of a complex independently because we used information of ligand and receptor gene expression in the cell types of interest to model cell-cell communication conveyed by proteins. We acknowledge that lowly expressed ligand or receptor genes may be translated and be functional at the protein level. The study shown in Blood 1996;87:2419-2427 that compares mRNA and protein expression of IL2RA in AML samples is an example. In 3 (out of 14) samples, IL2RA gene expression was not detectable, but IL2RA protein expression was detected on cell surface. To control for such possibility, we treated the connections between a ligand and the receptors of a hetero-multimeric complex independently in our network model. The connection between a ligand and a cell in our network indicates the probability that the cell will respond to the ligand stimulation, as oppose to that a cell will respond to the ligand stimulation. This has been emphasized in the manuscript, “Given that our network was constructed from gene expression data and it is possible that receptor proteins encoded by lowly expressed genes are functional for signal transduction (Schumann *et al*, 1996). From a modeling perspective, we assumed that the greater the number of receptor species that a cell expresses for a ligand, the higher the probability that the ligand binds to the cell. We considered the interactions of each ligand and its cognate receptors independently; the practice did not affect our conclusions on network structures as shown below” (page 7 lines 7 – 12).

Regarding the connections between IL7 and neutrophils / eosinophils, our network model suggested that neutrophils and eosinophils are potential responders to exogeneous IL7 stimulation, but this prediction needs to be experimental validated as we did for the HSCe-targeting ligands.

If I am interpreting the text correctly (pg 7), the approach taken by the authors would treat expression of IL7R and IL2RG as additive for response to IL7 but this makes no sense. Without both, the cell will not respond at all. In addition, neutrophils are described as responding to IL-2 due to their expression of IL2RG but they do not appear to express IL2RA or IL2RB.

Neutrophils did differentially over-express IL2RB gene as shown in Table E3, and the connection between IL2 and neutrophils via IL2RB is in Table E4. In addition, an *in vitro* study has shown that neutrophils respond to exogeneous IL2 stimulation at low affinity because of the presence of IL2RB but not IL2RA on cell surface (Blood. 1994;84:3870); the connection between IL2 and neutrophils in our network is valid.

As a final example, eosinophils are listed as responding to IL21 (again because of IL2RG) but they do not express IL21R, which would be required for this to be the case. The influence of cases such as these

would be to increase the apparent promiscuity of the ligand binding network which is exactly what the authors observe.

See above response regarding the connection between IL7 and neutrophils / eosinophils.

4. Figure 4F

As I interpret these graphs (and I might be wrong, they're not clear), each line represents the percentage of possible ligands with a particular function that are available to HSC-e cells given a varying probability that the ligands produced by a particular cell type are available to HSC-es. If this is the case, the value at $x=0$ is the number of ligands available if the particular cell type is not contributing anything. So, if it is high at $x=0$ and the line remains relatively flat then that cell type doesn't contribute many ligands no matter how "close" it is to the HSCe compartment. This appears to be the case for megakaryocyte apoptotic ligands (green line, top left graph); with zero probability of megakaryocytes contributing to the ligands, there are still ca. 66% of the set available to the HSC-es. This only increases to ca. 70% when $PMCN\text{-mega}=1$. It is hard to reconcile this with the statement in the text that says "Our simulation results revealed the importance of Mega ligands to HSC-e apoptosis".

The purpose of the analysis is to understand the effect of each cell type on HSCe fate as a function of the spatial relationship between the cell type of interest and HSCe. Therefore, for each cell type, we should look at the difference between the contribution score at $x = 1$ and the contribution score at $x = 0$. Presentation of the results of this analysis has been revised and described in the figure legend (page 43 line 14). Further, we redefined functional categories to increase the resolution of our analysis. The new results are shown in Figure 4G to replace Figure 4F in the initial submission. The results are presented on page 12 (lines 5 – 14) of the revised manuscript.

5. High-content screen

Text on page 14 states "At a working concentration of 100 ng/ml however the ligand led to a significant decrease in the number of HSC-enriched cells ($P\text{-value} = 0.0007$), progenitor cells ($P\text{-value} = 0.0094$) and mature cells ($P\text{-value} = 0.0207$) (Figure E7B-ii) so it inhibited HSC-e proliferation, which is consistent with its well-known pro-apoptotic effect." Inhibition of proliferation is very different from promotion of apoptosis. Was apoptosis observed after treatment with this ligand?

We did not perform a formal apoptosis assay. However, the phenotype of decreasing in cell numbers of all three compartments (HSC-enriched cells, progenitors and mature cells) is consistent with the phenotype of apoptosis. We clarified our interpretation of the data in the revised

manuscript, “At a working concentration of 100 ng/ml however the ligand led to a significant decrease in the number of HSC-enriched cells (ANOVA P-value = 0.0007), progenitor cells (ANOVA P-value = 0.0094) and mature cells (ANOVA P-value = 0.0207) (Figure E6B-ii) so it inhibited HSC-e proliferation, which may be due to the pro-apoptotic effect of the ligand (Zamai *et al*, 2000)” (page 15 lines 8 - 12).

In these data, TNFSF10 promotes self-renewal/proliferation at 10 ng/ml but is inhibitory (or maybe pro-apoptotic) at 100 ng/ml. In the Zamai (2000) paper cited, TNFSF10 causes measurable apoptosis of HL-60 cells at 10 ng/ml. Since HL-60 cells are mostly CD34- (Kuranda, K *et al*. 2011, *J. Cell. Biochem.*, 112: 1277-1285) one might expect non-transformed mature cells to behave in a similar way and to exhibit a reduction in numbers.

Do the authors think that these discrepancies are a reflection of differences between cell types or does it represent a lack of sensitivity in their assay?

Thank you for the question, but we are confused about the reviewer’s reference to HL-60 cells. In Zamai *et al*. (*Blood*. 2000, 95(12):3716) showed that 10µg/ml TNFSF10 induced apoptosis of peripheral blood-derived erythroid cells (not HL-60) after 20hr incubation.

For the reviewer’s information, we did see discrepancy between our results and published results for the same ligand. In general, at least two factors can contribute the observed discrepancies. First, we used much more enriched human HSC population to test the ligands of interest than most if not all the *in vitro* studies of hematopoietic stem and progenitor cells in literature. Second, our ligand usage is different from existing literature. See our response to this reviewer’s major point 7i for more details.

6. Data in figure 5C and Supplementary Table are presented as P values. To aid full interpretation of these results it would be helpful for the authors to show the magnitude of the change as well as its significance.

A direct comparison between the output of the basal cytokine condition and that of the testing conditions is included in Figure E8. A reference has been included in the revised manuscript, “See Figure E8 for cell number comparison between the tested conditions and the BC control” (page 14 lines 8 – 9).

7. The selection criteria for ligands for the high-content screen include using “literature anticipated HSC-targeted ligands”. Nearly half of the selected ligands (15/33) have a literature reference in Table E5. In some cases it seems that the results presented here simply recapitulate the existing data

and confirm that these ligands do, indeed, have an effect upon HSCs. It maybe, therefore, not surprising that there is a "significant enrichment of prediction capacity in this analysis". I would like to see the authors address this by commenting on i) whether their observations agree with published literature. ii) What the enrichment of prediction capacity if the already-published ligands are ignored. iii) Whether the published ligands would have been chosen by the other two criteria dependent solely upon the ligand-receptor pairs and transcriptomic data. This would strengthen the authors' claims that the high-throughput screen was truly a validation of the network.

(i) Whether their observations agree with published literature

Our observations do not fully agree with literature. For example, addition of BMP4 to cord blood $\text{Lin}^- \text{CD34}^+ \text{CD38}^-$ (containing approximately 1/617 SRC) maintained population of hematopoietic reconstituting cells (or HSC by definition) and increased $\text{CD34}^+ \text{CD38}^-$ progenitor cells, compared to the cultures without BMP4 addition. The study suggested that addition of BMP4 induced HSC differentiation, which is consistent with our observation about the role of BMP4 in directing HSC-e cell fate. However, our observation about VIP is inconsistent with the literature. Kawakami et al. (Leukemia. 2004, 18;5:912-921) found that addition of VIP at an optimal dose to cord blood $\text{Lin}^- \text{CD34}^+$ (containing approximately 1/147000 HSC) cell culture increased the total number of cells in 7 days in a single cell assay, and increased the numbers of primitive progenitors and definitive progenitors in clonal assays, comparing to the cultures without VIP addition. In our hands, the ligand was neutral to HSC-e (containing approximately 1/13 HSC).

At least two factors can contribute the discrepancy between our observation and the literature. First, we used much more enriched human HSC population to test the ligands of interest than most if not all the *in vitro* studies of hematopoietic stem and progenitor cells in literature. Most of the experiments in the existing literature in the field were performed on either Lin^- , CD34^+ or $\text{CD34}^+ \text{CD38}^-$ cells. Only one in every 147000 (for Lin^- and CD34^+) or 617 (for $\text{CD34}^+ \text{CD38}^-$) of these cells was HSC as defined by the transplantation assay. In comparison, approximately one in every 13 HSC-e that we used was HSC (Science.2011;333(6039):218-21). Second, our ligand usage is different from existing literature. In the hematopoietic field, it is known that ligands can have synergistic, antagonistic or additive effects on cell fate decisions. Difference in ligand usage in terms of ligand identity and concentration can make a difference to culture output. To sum up, we included the following sentences in the revised Discussion,

“Discrepancy between our observation about the *in vitro* effects of the tested ligands and their documented effects in literature may be attributable to the differences in experimenting cell populations and culture conditions” (page 21 lines 17 – 19).

- (ii) Whether the published ligands would have been chosen by the other two criteria dependent solely upon the ligand-receptor pairs and transcriptomic data.

The ligands would have been chosen without a literature survey. There are 117 HSCe-targeting ligands in the cell-cell communication network as we mentioned in line 21 on page 7 of the manuscript. We compared their receptor expression levels in HSCe to the other cell types and checked the molecular interaction confidence scores of the ligand-receptor pairs (Figure E4). Ideally, we would test all the ligands which receptor expression levels were evidentially high in HSCe for their activity on HSC-e (isolated using Rho^{low} in addition to the cell surface marker of HSCe). However, the HSC-e population is rare in cord blood samples. We had to prioritize the ligands for experiments. Thus, we performed a literature survey on ligands that had been used in *in vitro* cell culture of human cord blood-derived hematopoietic stem and progenitor cells; 11 ligands fell in this category. Ligands such as ANGPT1, ANGPT2, ANGPTL3 and BMP2 had been used in mice or human bone marrow cells, so they were also prioritized for experiments in our study.

We have clarified our strategy for ligand selection in the following sentences, “HSCe-targeting ligands in the CCC network (Table E4) were ranked according to the molecular interaction confidence scores (Ceol *et al*, 2010) for ligand-receptor interactions (Table E2) and the receptor gene expression levels in HSCe from the transcriptome data. Thirty-three ligands were prioritized for experimental tests (Materials and Methods, Table E7)” (page 13 lines 4 – 7), and in Materials and Methods (page 30 lines 7 – 12).

- (iii) What the enrichment of prediction capacity if the already-published ligands are ignored

We calculated the enrichment of prediction capacity after removing the 15 ligands with references in Table E7. Assuming that the probability that a randomly selected ligand is functional is 0.5, and that the tested ligands are independent from each other, the probability of having 15 effective ligands is 0.003 (see below calculation). This result has been included in the Materials and Methods (page 30 lines 12 – 14), and referenced in the revised

manuscript (page 15 line 2).

$$P(X = 27) = \binom{18}{15} 0.5^{15} (1-0.5)^3 \approx 0.003$$

8. Discussion, pg17

It is unclear which data support the claim that "...the mature cells, particularly mono and granulocytes (neut, baso and eos) were found to mainly express inductive ligands for HSC-e proliferation and differentiation." Figure 6B indicates that monocytes are enriched for production of self-renewal inducing and proliferation inhibiting ligands; neutrophils for the same as well as quiescence inducing; eosinophils are only enriched for proliferation inhibiting ligands.

The typo has been corrected. It should read "...were found to mainly express inhibitory ligands for HSC-e proliferation and inductive ligands for differentiation" (see page 19 line 8).

Minor points

1. Figure 2B

Shading for leukocyte chemotaxis gene expression in monocytes indicates a Z-score > 1.15 but no Z-score is given for this in Table E1. Likewise, no Z-score is present in Table E1 for PREB differentiation genes in PreB cells. Generally, comparison between figure 2B and Table E1 is impeded by the inconsistent ordering of cell types and gene types between the two tables.

Table S1 has been updated and formatted as Figure 2B.

2. Figure 6Ciii

The weighted network here is based on cell numbers for the mono-nucleated cell compartment. However, in figure 4Ci, the authors assert that due to cell numbers in this compartment, HSCes have a "negligible probability of accessing ligand resources in the system." Please could the authors explain the purpose of illustrating their network with this situation? Is it to emphasise the role of spatial compartmentalisation? If so, this could be more clearly discussed.

We apologize for the unclear description of the purpose of the analysis. As the reviewer interpreted, we want to emphasize the effect of spatial compartmentalization on HSC fate regulation through feedback signaling. This spatial effect is more evident when we look at the hematopoietic system as a whole where HSC reside in the bone marrow niche, and mature cells mainly circulate in the peripheral blood / tissues. In our study, we weighted the HSC feedback network in Figure 7C (Figure 6Cii in the manuscript of the initial submission) by the cell composition in mono-

nucleated cell compartment which approximates the cell composition of the whole hematopoietic system, and then we imposed another weighting parameter to model the effect of the physical barrier between the HSC niche and the peripheral blood / tissue *in vivo*. To address the reviewer's comment, we clarified the purpose of this analysis in the following sentence, "...to illustrate the roles of cellular dynamics and spatial distribution in HSC fate regulation through feedback signaling" (page 17 lines 14 – 15). Legend of Figure 7 (restructured from Figure 6C in the initial submission) was written to explain of our analysis in details.

3. Figure E2 legend
"Using the height of Fithe primitive cell module...". Assume means "using the height of the..."

Corrected

4. Figure E4 legend
"...setting the reference for cauterising the effects..." surely should be "...characterising the effects..."

Corrected. The old Figure E4 has been renumbered as Figure E6 in the revised manuscript.

5. Figure E5b
I found the use of circles of different sizes to represent cell numbers vs control hard to interpret. What is scaled relative to cell number? Radius? Area? Circumference? This figure can be easily interpreted as a two-dimensional measure (area) changing to represent a one-dimensional quantity (cell number). What's wrong with a bar chart to represent this?

Thank you for the comment and suggestion. We think that bar charts will mislead the readers regarding the statistical test that we used. Side-by-side comparison of two bars (in this case, a control condition and a test condition) implies direct comparison between two conditions using statistical tests such as t-test. However, we did not do pairwise comparison but used nested ANOVA to take into account all the test conditions at once. Therefore, we used circles of different sizes to represent differences in cell numbers between the test conditions and the control. We agree that it is not clear which parameter of a circle that we use to represent cell number, but we think that radius, area and circumference all convey the same message in this case. To further address the reviewer's comment, we included a sentence to the legend of Figure E7 (renumbered from Figure E5b in the initial submission), "The schematics next to the flow cytometry data represent differences in cell numbers (by circle sizes) between test conditions and the control condition" (page 53 lines 6 – 7).

6. Table E2: Poorly formatted. The final column is split onto a separate page.

Corrected

7. Table E3: Receptors overexpressed in eosinophils are not listed

Corrected

8. Table E7: What are the measurements of cell numbers? Is it $\times 10^3$?

The old Table E7 is renumbered as Table E8 in the revised manuscript. Numbers shown in table are absolute cell numbers.

9. Table E8: Poorly formatted. The final column is split onto a separate page.

Corrected

10. Page 6: "Thus, we suspected the existence of gradients of receptor and ligand expression that paralleled to the hematopoietic developmental hierarchy (Figure 2A)." Gradient is not a very helpful word here. When talking about cell-cell signalling I think of spatial concentration gradients. I think the authors mean "changes in the receptors/ligands expressed during progression through differentiation". A different word would help.

Corrected

11. Page 8: "...supported the existence of 3 ligand-cell modules". Four modules are then listed.

Corrected

12. Page 9: "...relative cell frequency to allow more abundant cell types skew the ligand resources...". I think there's a word missing/extra/out of order here.

Corrected. It should read "relative cell frequency that allows more abundant cell types skew the ligand species and resources available to HSCe" (page 10 lines 7 – 8).

13. Page 15: Several typographic errors. Please correct them.

Corrected

Thank you again for submitting your work to Molecular Systems Biology. We have now heard back from the two referees who were asked to evaluate your manuscript. As you will see from the reports below, while the main concerns of reviewer #1 have been satisfactorily addressed, reviewer #3 mentions two issues that we would ask you to address in a revision of the manuscript. These points refer to the need to provide additional comments/clarifications regarding i) the potential effect of hetero-multimeric receptors and ii) the p values related to the data shown in Figure S8.

Reviewer #1:

The authors have done a strong job of addressing the prior comments, in particular by adding more biological insight about the ligands they have studied and the cell types they are associated with. They have also supported their model findings with some high throughput in vitro work, which strengthens the paper. This is a high quality manuscript.

Comments on revision responses to prior major points:

1. General remark 3: Was anything new actually found? The authors do a much better job in this draft explaining their specific contributions.
2. General Remark 5/Major Point 2: The need to add more biological insight to the network maps. The authors successfully addressed the prior comment with Figure 7 A,B (which I think are very effective!), Figure E9, and added text.
3. Major Point 3: Network rigidity. I commend the authors for their rigor in assessing the rigidity of their network and ligand binding promiscuity, as well as adding a line to their discussion about limitations in their method, having not used temporal or condition specific data. While I still do not think that their conclusions are as strong/meaningful as could be if they used temporal/conditional specific data, at least it is acknowledged here.

Sole additional (minor) comment:

1. Use of the term "design principles" as the goal of the project. I think this term is a tad misleading/difficult to understand. Authors should simply say that they are trying to uncover the mechanisms of cell-to-cell communication by which hematopoietic cells influence hematopoietic stem cell fate.

Reviewer #3:

Having read the authors' rebuttal letter I have two main concerns relating to my original comments. These are detailed below and are followed by a discussion of all of the points raised by all three reviewers.

Main issues from Reviewer 3

Major point 3

I still find the way the authors have approached the question of heteromultimeric receptors problematic.

1) The authors show an adjusted plot for the ligand production network if heteromultimeric receptors are taken into account (figure E2C). It shows a different number of ligands than the plot in figure 3A. Since "A cell-to-ligand interaction, I_{ij} , in the ligand production network was defined if cell i produced ligand j " I fail to see how changing the definition of a receptor will alter the ligand production network. I'd appreciate a comment from the authors to clarify this.

2) "From a modeling perspective, we assumed that the greater the number of receptor species that a

cell expresses for a ligand, the higher the probability that the ligand binds to the cell." - This statement would be acceptable if all members of a heteromultimeric receptor only associated with a single ligand. However, this is not the case. For example, the common gamma chain (IL2RG) is associated with receptors for at least six different ligands. It wouldn't take too many proteins such as this to dramatically increase the apparent promiscuity of a ligand binding network. If the authors wish to state with confidence that the ligand binding network is more promiscuous, they really should address this point more thoroughly.

3) Related to point 2 is the authors' statement that "lowly expressed ligand or receptor genes may be translated and be functional at the protein level. The study shown in Blood 1996;87:2419-2427 that compares mRNA and protein expression of IL2RA in AML samples is an example. In 3 (out of 14) samples, IL2RA gene expression was not detectable, but IL2RA protein expression was detected on cell surface." Having read the Blood paper to which the authors refer, this statement is troubling.

I am concerned that the claim that "In 3 (out of 14) samples, IL2RA gene expression was not detectable, but IL2RA protein expression was detected on cell surface." misrepresents the findings in the referred to paper. Schumann et al. show that IL2RA gene expression was detectable in eight out of 14 AML samples but that only two samples had detectable protein levels. Both of these samples had detectable mRNA expression. I presume that the authors of the paper under review are interpreting samples 3, 4 and 13 as "positive" for IL2RA protein expression even though they are negative for mRNA. These three samples show 2%, 3% and 1% of cells being IL2RA positive by flow cytometry. I agree with Schumann et al. that these cells should not be considered positive for IL2RA expression when thinking about their signaling response. It should also be noted that Scatchard analysis of one of the samples (AML-4) did not indicate the presence of high- or medium-affinity binding sites for IL-2.

Thus, I do not agree that the Schumann paper in any way indicates that receptors without detectable mRNA may still be functional in signalling.

4) Even if we were to accept that signalling molecules without detectable transcription were available for use in signalling, this would still be problematic for general acceptance of the authors' methods in constructing the signalling networks. Everywhere else in the paper, the authors use expression levels from transcriptomic data to define the functionality of their cells. It causes cognitive dissonance to hold this idea in one's head at the same time as accepting that genes without detected expression could be producing functional protein. Should we accept that monomeric receptors are also available for signal transduction even if their expression is not detected? What about "non-expressed" ligands?

5) The authors claim that "The promiscuous network structure is robust...[to] the incorporation of heteromultimeric receptors" and illustrate this with figure E3C. From inspection of this figure and comparison with figure 4A it appears that the cells are less promiscuous. In particular, HSCe no longer appear to respond to the highest number of ligands which disagrees with the conclusion in the text that these are the most promiscuous cell type. I would like to see a quantitative measure of "promiscuity" rather than relying solely on inspection of the spectral co-clustering plots.

6) My apologies for missing the inclusion of IL2RB in the Neutrophil list in table 3B.

Major point 6

Thank you for including Figure E8. However, it is, at times, very difficult to reconcile the calculated p values with the data shown in Figure E8. For example, it is not at all obvious why, amongst others, TNFSF12, FGF2, IL12A, BMP2 and NGF are assigned p values suggestive of significant increase in HSCe numbers compared with the control. Please could the authors check their statistical tests? If these counterintuitive p values are due to the correction of batch effects (by including experiment date in the ANOVA) it would be valuable to include versions of E8 that separate out the different batches.

Alternatively, is this due to not correcting for multiple testing? I can appreciate the desire to avoid missing effective ligands if using lists to inspire further research. However, I think that in drawing biological conclusions and generating networks it may be valuable to use the corrected P values.

Reviewer 1's comments

General remarks 3, 4 & 5

The authors address Reviewer 1's comments on impact and novelty by disagreeing with them and emphasising their key findings and how they "extract design principles of the cell-to-cell communication network", "summarize a design principle of the haematopoietic system" and that their work will be "applicable to studies of other multicellular systems".

I am inclined to agree with Reviewer 1 that this is a systematic and rigorous demonstration of things that were already broadly known in the field. However, this does not mean that it is without value since, as the authors point out, it is through analyses such as these that we will be able to understand cell-to-cell signalling networks. Resolution of this disagreement is best left to the editor.

General remark 5

The new work performed to show pathway enrichment (Fig 7) is appreciated and does draw more biological relevance from the data. However, not all of it is novel since, for example, NF κ B is already known to play a role in HSC self-renewal (<http://www.ncbi.nlm.nih.gov/pmc/articles/PMC3602314/>). I would like to see the authors provide more discussion to set these findings into the context of what is already known so that it is clear what is truly novel and what is essentially (useful) validation.

Major point 1

The new figure 3C certainly provides what Reviewer 1 requested in terms of expanding the functional categories. However, I'm not sure that it shows more convincing functional groups nor any difference from Reviewer 1's original comment that "Almost all of the categories display almost all of the functions just at slightly varying levels".

Major point 2

Breaking down the biological functions into more categories does address Reviewer 1's point to an extent although it remains very descriptive. A discussion of how the differential expression between these categories relates to the biology of haematopoiesis would help here.

Major point 3

I don't think that a comparison between the ex vivo data in the paper and cultured megakaryocytes addresses the Reviewer's comments. I do, however, appreciate the authors' addition to the discussion that highlights this limitation.

I agree that the ligand binding does still appear to be promiscuous without HSCs but please see my later comment about lack of quantitation of promiscuity.

Major comments 4 and 5

The authors' responses address the reviewer's concerns and I appreciate the addition of the discussion of the study's limitations.

Minor comment 1

Please see comments about Reviewer 3's major point 2 below.

Reviewer 2's comments

Point 1

I am broadly happy that the authors address this concern within the scope of their study. Are the authors content that the change in magnitude between the phenotypic and functional response to IL11 and CSF2 in fig E5/R2.1 is not important?

Points 2-4

The authors responses and changes to their text are fine.

Reviewer 3's comments

Major point 1

I appreciate the authors' explanation of their choice of methodology here. One concern remains.

How robust are their conclusions to changing the "arbitrary cut-off" of six cell types when defining overexpression? They go to great lengths to show robustness to different FDRs and equivalent reassurance about robustness to this parameter would be helpful. Since the choice of differentially expressed genes underpins the rest of the work it is important.

Major point 2

I now understand the authors' motivation for choosing this FDR but I would suggest that they find a different way to illustrate it. In a standard assessment of a classifier, anything with equal true positive and false positive rates is the same as randomly assigning classes and is, essentially, useless.

Here, the problem is that the lists of genes that the authors use for comparison do not represent an accurate "ground truth" and so cannot be used to accurately calculate these rates. The lists are not comprehensive and nor does exclusion from the list imply that a gene has no relevance (if both of these were the case then the lists alone would suffice for network construction and assessment of differential expression would not be necessary!). I think that a change in presentation and terminology (away from "true positive"/ "false positive") would help to make this clearer.

Major point 4

Rescaling these graphs certainly makes them look more convincing. I would be happier to see these data also presented as in the original graphs so that the absolute values can also be observed alongside the rescaled values.

Major point 5

Apologies for referring to the incorrect paper with HL-60 data. I meant to refer to <http://bloodjournal.hematologylibrary.org/content/100/7/2421.full.pdf> which does discuss the pro-apoptotic response of HL-60 cells to TNFSF10.

I appreciate the authors' discussion about the discrepancies between their data and other studies.

Major point 7

If the concerns about p values (discussed at the start) can be satisfactorily addressed then the authors' response to this point is fine. I particularly appreciate the quantification of enrichment if the previously reported ligands are removed.

(see next page)

Reviewer 1

The authors have done a strong job of addressing the prior comments, in particular by adding more biological insight about the ligands they have studied and the cell types they are associated with. They have also supported their model findings with some high throughput in vitro work, which strengthens the paper. This is a high quality manuscript.

Comments on revision responses to prior major points:

1. General remark 3: Was anything new actually found? The authors do a much better job in this draft explaining their specific contributions.
2. General Remark 5/Major Point 2: The need to add more biological insight to the network maps. The authors successfully addressed the prior comment with Figure 7 A,B (which I think are very effective!), Figure E9, and added text.
3. Major Point 3: Network rigidity. I commend the authors for their rigor in assessing the rigidity of their network and ligand binding promiscuity, as well as adding a line to their discussion about limitations in their method, having not used temporal or condition specific data. While I still do not think that their conclusions are as strong/meaningful as could be if they used temporal/conditional specific data, at least it is acknowledged here.

Thank you.

Sole additional (minor) comment:

1. Use of the term "design principles" as the goal of the project. I think this term is a tad misleading/difficult to understand. Authors should simply say that they are trying to uncover the mechanisms of cell-to-cell communication by which hematopoietic cells influence hematopoietic stem cell fate.

Thank you for this comment. Certainly, we agree that although our goal is to understand the "design principles" of human hematopoiesis, our paper is focused on uncovering mechanisms of cell-to-cell communication and the feedback regulation of hematopoietic stem cell fate. We have thus revised our language throughout the manuscript. Specifically,

- Page 2 lines 14 – 15 (Abstract): "This study uncovers cellular mechanisms of hematopoietic cell feedback in HSC fate regulation, provides insight into the design principles of the human hematopoietic system";
- Page 5 lines 1 – 3: "Overall, our findings provide insight into the design principles of the human hematopoietic system focusing on the mechanisms of CCC in the feedback regulation of HSC fate";
- Page 20 lines 15 – 16: "In summary, our results demonstrate the importance of cell-to-cell communication in human UCB stem cell fate control".

Major point 3: I still find the way the authors have approached the question of heteromultimeric receptors problematic.

- 1) The authors show an adjusted plot for the ligand production network if heteromultimeric receptors are taken into account (figure E2C). It shows a different number of ligands than the plot in figure 3A. Since "A cell-to-ligand interaction, I_{ij} , in the ligand production network was defined if cell i produced ligand j " I fail to see how changing the definition of a receptor will alter the ligand production network. I'd appreciate a comment from the authors to clarify this.

Changing the definition of a receptor can affect the ligand production network for the following reason. When taking heteromultimeric receptor expression into account during network construction, some ligands in our original cell-cell communication (CCC) network do not have compatible receptor(s) (i.e., all subunits required for binding) expressed on the 12 cell types analyzed, and therefore the ligands are "orphan" and cannot be included in the heteromultimeric receptor-based CCC network. Since the ligand production network was extracted from the CCC network, the number of ligands in the ligand production network tends to decrease as we consider heteromultimeric receptor expression. To make the construction method for ligand binding networks clear, we have added the following sentence to the Results section (page 9 lines 18 – 19), "a ligand-to-cell interaction, B_{ji} , in the ligand binding network was defined if cell i expressed receptor(s) for ligand j ".

We appreciate careful examination that the reviewer has given our results. We found a small error in our code for constructing the heteromultimeric receptor-based CCC network in which a ligand-to-cell interaction only occurs when both the ligand binding arm and the signaling arm of a heteromultimeric receptor are differentially over-expressed. By fixing the error, the heteromultimeric receptor-based CCC network contains 172 ligands, compared to 178 in the original network. As justified above, ligands OSM (produced by GMP and EryB), IL7 (produced by MEP, Mega and Baso), IL15 (produced by Neut and Mono), IL3 (produced by CMP), LIF (produced by Neut and EryB), and CTF1 (produced by EryB) are not in the heteromultimeric receptor-based CCC network. The result of the ligand production network (Figure E2C) and that of the ligand binding network (Figure E3C) have been corrected. The six ligands (OSM, IL7, IL15, IL3, LIF and CTF1) that are affected by modeling heteromultimeric receptor expression are listed in the legend of Figure E2C. In addition, we have indicated interactions involving heteromultimeric receptors in Table E4 that shows the content of our CCC network.

- 2) "From a modeling perspective, we assumed that the greater the number of receptor species that a cell expresses for a ligand, the higher the probability that the ligand binds to the cell." - This statement would be acceptable if all members of a heteromultimeric receptor only associated with a single ligand. However, this

is not the case. For example, the common gamma chain (IL2RG) is associated with receptors for at least six different ligands. It wouldn't take too many proteins such as this to dramatically increase the apparent promiscuity of a ligand binding network. If the authors wish to state with confidence that the ligand binding network is more promiscuous, they really should address this point more thoroughly.

We interpret the reviewer's comment as following. For heteromultimeric receptors of the same family, the signaling arm of the receptors is shared and transmits the signaling activities of multiple ligands. For example, CSF2R β transmits CSF2, IL3 and IL5 signals; IL6ST (gp130) transmits IL6, IL11, LIF, OSM, CTF1 and CNTF signals; and IL2RG transmits IL2, IL4, IL7, IL9, IL15 and IL21 signals. Given our original network construction protocol that treats monomeric receptors independently, if a cell type expresses IL2RG (for example), 6 ligands would bind to the cell type. According to the reviewer, this artificially increases the promiscuity of the ligand binding network. We agree with the reviewer's concern. However, such case only represents 23 interactions in our original ligand binding network, and many cell types express both arms of heteromultimer receptors (Table R1). Thus, we did not expect adding the concepts of heteromultimeric receptor binding to affect the promiscuous structure of the ligand binding network.

To quantify our prediction, we constructed CCC networks that model heteromultimeric receptors to different degrees of detail:

- (1) Treat the interactions between a ligand and monomeric receptors independently, i.e., our original method;
- (2) Establish the interaction between a ligand and a cell if the ligand binding arm of a heteromultimeric receptor is expressed;
- (3) Establish the interaction between a ligand and a cell only if both the ligand binding arm(s) and the signaling arm of a heteromultimeric receptor are expressed.

Then, we compared the degree distributions in the ligand binding network and in the ligand production network. As shown in Figure R1, regardless the models used, HSCe bound to the largest number of ligands. In addition, ligands in the production networks interacted with at most 5 cell types, whereas certain ligands in the binding networks interacted with up to 12 cell types, implying that the ligand-cell interactions in the binding networks were less modular (i.e., more promiscuous) than the interactions in the production networks. These results have been included and discussed in Figure E3.

Table R1. Ligands that are affected by heteromultimetric receptor modeling. The table shows heteromultimeric receptor expression in the 12 cell types analyzed. There are 23 interactions that only involve signaling arms of heteromultimeric receptors; 15 interactions that only involve binding arms of heteromultimeric receptors; and 9 interactions of heteromultimeric receptors.

| Ligand | Receptor | HSCe | CMP | GMP | MEP | MLP | Mega | EryB | Neut | Baso | Eos | Mono | PreB |
|---|---|---|---|---|---|---|---|---|---|---|---|---|---|
| | binding arm | | | | | | IL2RA | | | IL2RA | | | |
| IL2 | signaling arm | | | | | | | | | | IL2RG | | |
| | multimer | | | | | | | | IL2RB:IL2RG | | | | |
| | binding arm | | | IL3RA | | IL3RA | | | | | | | |
| IL3 | signaling arm | | | | | | | | CSF2RB | | CSF2RB | CSF2RB | |
| | multimer | | | | | | | | | | | | |
| | binding arm | | IL5RA | | IL5RA | | | | | | | | |
| IL5 | signaling arm | | | | | | | | CSF3RB | | | CSF2RB | |
| | multimer | | | | | | | | | | IL5RA:CSF3RB | | |
| | binding arm | | | | | | | | | | | IL6R | |
| IL6 | signaling arm | IL6ST | | | | | | IL6ST | | | | | |
| | multimer | | | | | IL6R:IL6ST | | | | | | | |
| | binding arm | | | | | | IL7R | | | | | | |
| IL7 | signaling arm | | | | | | | | IL2RG | | IL2RG | | |
| | multimer | | | | | | | | | | | | |
| | binding arm | | | | | | | | IL11R | | | | |
| IL11 | signaling arm | | | | | | | IL6ST | | | | | |
| | multimer | IL11RA:IL6ST | | | | IL11RA:IL6ST | | | | | | | |
| | binding arm | | | | | | | | | | IL13RA1 | | |
| IL13 | signaling arm | | | | | | | | | | | | |
| | multimer | | | | | | | | IL4R:IL13RA1 | IL4R:IL13RA1 | | IL4R:IL13RA1 | |
| | binding arm | | | | | | | | IL2RB | | | | |
| IL15 | signaling arm | | | | | | | | IL2RG | | IL2RG | | |
| | multimer | | | | | | | | | | | | |
| | binding arm | | | | | | | | | | | | |
| IL21 | signaling arm | | | | | | | | | | IL2RG | | |
| | multimer | | | | | | | | IL21R:IL2RG | | | | |
| | binding arm | | | | | | | | | | | | LIFR |
| LIF | signaling arm | IL6ST | | | | IL6ST | | IL6ST | | | | | |
| | multimer | | | | | | | | | | | | |
| | binding arm | | OSMR | | | | | | | | | | LIFR |
| OSM | signaling arm | IL6ST | | | | IL6ST | | IL6ST | | | | | |
| | multimer | | | | | | | | | | | | |
| | binding arm | | | | | | | | | | | | LIFR |
| CTF1 | signaling arm | IL6ST | | | | IL6ST | | IL6ST | | | | | |
| | multimer | | | | | | | | | | | | |

A. Monomeric receptor-based network

B. Receptor binding arm-based network

C. Heteromultimeric receptor-based network

Figure R1. Structure comparison between networks that modeled heteromultimeric receptors at three degrees of detail.

A. Network that treats the interactions between a ligand and monomeric receptors independently. (i) Spectral co-clustered ligand binding network. The numbers on the right indicate the number of bound ligands of each cell type. (ii) The numbers of interacted cell types of each ligand in the ligand binding network and in the ligand production network.

B. Network that counts ligand-to-cell interactions when the ligand binding arm of a heteromultimeric receptor are expressed. (i) Spectral co-clustered ligand binding network. (ii) The numbers of interacted cell types of each ligand in the ligand binding network and in the ligand production network.

C. Network that count ligand-to-cell interactions when both the binding arm and the signaling arm of a heteromultimeric receptor are expressed. (i) Spectral co-clustered ligand binding network. (ii) The numbers of interacted cell types of each ligand in the ligand binding network and in the ligand production network.

- 3) Related to point 2 is the authors' statement that "lowly expressed ligand or receptor genes may be translated and be functional at the protein level.

The study shown in Blood 1996;87:2419-2427 that compares mRNA and protein expression of IL2RA in AML samples is an example. In 3 (out of 14) samples, IL2RA gene expression was not detectable, but IL2RA protein expression was detected on cell surface." Having read the Blood paper to which the authors refer, this statement is troubling.

I am concerned that the claim that "In 3 (out of 14) samples, IL2RA gene expression was not detectable, but IL2RA protein expression was detected on cell surface." misrepresents the findings in the referred to paper. Schumann et al. show that IL2RA gene expression was detectable in eight out of 14 AML samples but that only two samples had detectable protein levels. Both of these samples had detectable mRNA expression. I presume that the authors of the paper under review are interpreting samples 3, 4 and 13 as "positive" for IL2RA protein expression even though they are negative for mRNA. These three samples show 2%, 3% and 1% of cells being IL2RA positive by flow cytometry. I agree with Schumann et al. that these cells should not be considered positive for IL2RA expression when thinking about their signaling response. It should also be noted that Scatchard analysis of one of the samples (AML-4) did not indicate the presence of high- or medium-affinity binding sites for IL-2. Thus, I do not agree that the Schumann paper in any way indicates that receptors without detectable mRNA may still be functional in signalling.

Even if we were to accept that signalling molecules without detectable transcription were available for use in signalling, this would still be problematic for general acceptance of the authors' methods in constructing the signalling networks. Everywhere else in the paper, the authors use expression levels from transcriptomic data to define the functionality of their cells. It causes cognitive dissonance to hold this idea in one's head at the same time as accepting that genes without detected expression could be producing functional protein. Should we accept that monomeric receptors are also available for signal transduction even if their expression is not detected? What about "non-expressed" ligands?

We thank the reviewer for the comments. The goal of our study was to extract design principles of the cell-cell communication networks in the normal hematopoietic system and to gain in-depth understanding of feedback regulation of HSC fate. As detailed in our response to this reviewer's point 2, adding the concept of heteromultimeric receptor binding did not affect our conclusions about the structures of the ligand production and binding networks. To address the reviewer's comment, we have explicitly identified the ligands that will be affected by modeling heteromultimeric receptor expression, and the ligands are listed in the legend of Figure E2C. To further address the reviewer's comment, we have deleted the following sentence from our manuscript, "and it is

possible that receptor proteins encoded by lowly expressed genes are functional for signal transduction (Schumann *et al*, 1996)".

- 4) The authors claim that "The promiscuous network structure is robust...[to] the incorporation of heteromultimeric receptors" and illustrate this with figure E3C. From inspection of this figure and comparison with figure 4A it appears that the cells are less promiscuous. In particular, HSCes no longer appear to respond to the highest number of ligands which disagrees with the conclusion in the text that these are the most promiscuous cell type. I would like to see a quantitative measure of "promiscuity" rather than relying solely on inspection of the spectral co-clustering plots.

As detailed in our response to the reviewer's point 1, we found a small error in our code for constructing the heteromultimeric receptor-based CCC network. After correcting the error, HSCe indeed bind to the highest number of ligands (Figure R1). Regarding the measure of promiscuity, see the results of degree analysis in our response to the reviewer's point 2.

- 5) My apologies for missing the inclusion of IL2RB in the Neutrophil list in table 3B.

Okay.

Major point 6: Thank you for including Figure E8. However, it is, at times, very difficult to reconcile the calculated p values with the data shown in Figure E8. For example, it is not at all obvious why, amongst others, TNFSF12, FGF2, IL12A, BMP2 and NGF are assigned p values suggestive of significant increase in HSCe numbers compared with the control. Please could the authors check their statistical tests? If these counterintuitive p values are due to the correction of batch effects (by including experiment date in the ANOVA) it would be valuable to include versions of E8 that separate out the different batches.

Alternatively, is this due to not correcting for multiple testing? I can appreciate the desire to avoid missing effective ligands if using lists to inspire further research. However, I think that in drawing biological conclusions and generating networks it may be valuable to use the corrected P values.

We apologize for our unclear description of the data shown in Figure E8. As the reviewer may have imagined, there was significant variability from experiment to experiment (so called batch effect) (Figure R2). While this is not unusual with primary cell culture, the batch effect confounded the comparison between test conditions and the control condition across experiments. We "corrected" the batch effect by treating "experiment ID" as the random factor in a mixed-effect

linear model. To clarify our method, we have included the following sentence in the Methods and Materials (page 29 lines 16 – 17): “a mixed-linear model was constructed with the experiment identifier as the random effect to account for the variability from experiment to experiment”.

The control averages shown in the original Figure E8 are the global average of all the experiments on the log10 scale. This does not truly reflect the control data used for the statistical test of each ligand. We have replaced the original Figure E8 with a plot that separates out the different batches. For each experiment, data of test conditions were normalized to data of the basal cytokine control. To address the reviewer’s question about TNFSF12, FGF2, IL12A, BMP2 and NGF, data for those ligands are shown in Figure R3. It is evident that addition of these ligands into cell culture increased the output of HSC-enriched cells ($CD34^+CD90^+CD133^+$). To make our analysis transparent, we have submitted the R script with the manuscript.

Figure R2. Variation in *in vitro* experiments. $CD34^+CD90^+CD133^+$ cell numbers, progenitor cell number and mature cell numbers from the basic control condition were compared across 33 *in vitro* experiments performed for this manuscript. Each experiment has three technical replicates. The average of all the samples are shown in red.

Figure R3. Comparison between the results of test conditions (TNFSF12, FGF2, IL12, BMP2 and NGF) and the basic control condition. The data of the test conditions were normalized to the data of the basic control condition in the same experiment. The experiment identifications are shown on the x-axis.